# Synthetic neuromorphic computing in living cells

Luna Rizik[1,5], Loai Danial[2,5], Mouna Habib[1,5], Ron Weiss [3,4] & Ramez Daniel [1] ✉

Computational properties of neuronal networks have been applied to computing systems using simplified models comprising repeated connected nodes, e.g., perceptrons, with decision-making capabilities and flexible weighted links. Analogously to their revolutionary impact on computing, neuro-inspired models can transform synthetic gene circuit design in a manner that is reliable, efficient in resource utilization, and readily reconfigurable for different tasks. To this end, we introduce the perceptgene, a perceptron that computes in the logarithmic domain, which enables efficient implementation of artificial neural networks in *Escherichia coli* cells. We successfully modify perceptgene parameters to create devices that encode a minimum, maximum, and average of analog inputs. With these devices, we create multi-layer perceptgene circuits that compute a soft majority function, perform an analog-to-digital conversion, and implement a ternary switch. We also create a programmable perceptgene circuit whose computation can be modified from OR to AND logic using small molecule induction. Finally, we show that our approach enables circuit optimization via artificial intelligence algorithms.

A central goal of synthetic biology[1–9] is to create large-scale genetic networks in living cells that implement sophisticated sensing, processing, and actuation[10–14]. To date, both the digital and analog computing paradigms have been implemented in living cells in an attempt to design and build genetic circuits efficiently. The digital paradigm, which abstractly computes with two discrete binary-coded levels [0,1][15], has inspired implementation of wide variety of genetic circuits, including logic gates[16–18], memory elements[19–21], a counter[22], state machines[23], a toggle switch[24], a digitizer[25], and highly complex logic functions[26,27]. The analog paradigm, in contrast, computes on a continuous set of numbers and has been suggested as an alternative to the digital paradigm for tasks that don't require decision-making[28–30]. Efforts in synthetic biology have also focused on other aspects of circuit control, such as complex temporal dynamics[31–33], and integral feedback controllers for robust adaptation[34,35].

Despite the many successful accomplishments to-date, significant challenges hinder further scaling of synthetic biological systems based on digital and analog computing[1,36]. Critical impediments include cellular resource limitations, high levels of random fluctuations, and undesirable interactions between synthetic parts and host cells[1,36,37]. Furthermore, digital design is often not suitable for computing with graded biological signals, while analog circuits may accumulate noise as they scale in size[37].

Alternatively, biological systems in nature exhibit nonlinearity across scales from the molecular level to network and inter-cellular systems and use redundant regulation and collective interactions to robustly execute highly sophisticated tasks; such as cell differentiation[38]. Furthermore, several theoretical analyses of certain gene regulatory networks demonstrate neural-like computational behavior[39–42]. Therefore, we have sought to adapt non-linear models in the form of neural-like computing[43,44] into individual single cells to overcome the aforementioned bio-design challenges.

The neuromorphic computing paradigm, which employs design principles and approaches of neuronal systems, has been successfully

[1]Department of Biomedical Engineering, Technion - Israel Institute of Technology, Haifa 3200003, Israel. [2]Andrew and Erna Viterbi Faculty of Electrical Engineering, Technion - Israel Institute of Technology, Haifa 3200003, Israel. [3]Synthetic Biology Center, Massachusetts Institute of Technology, Cambridge, MA, USA. [4]Department of Biological Engineering, Massachusetts Institute of Technology, Cambridge, MA, USA. [5]These authors contributed equally: Luna Rizik, Loai Danial, Mouna Habib. ✉e-mail: ramizda@bm.technion.ac.il

applied to a wide range of fields, including electronics[45–48], optics[49], software algorithms[44], and even in vitro DNA computing[50], leading to the realization of artificial intelligence systems. Neuromorphic systems efficiently solve complex tasks such as content addressable memory, pattern classification, object recognition, and optimization through machine learning algorithms. Furthermore, an advantage of neuromorphic computing systems compared to their digital counterparts is that implementing a given task often requires fewer computational devices[46,48], significantly when resources are scarce (such as synthetic biology). Neuromorphic systems usually combine analog information processing with decision-making capabilities using non-linear activation functions (e.g., sigmoid, rectifiers, step function) and support iterative optimization strategies. In these optimization strategies, characterization of circuit behavior is interleaved with iterative changes in computing device parameters, for example, based on prediction of how changes under consideration correlate with the derivative of an overall score function.

## Results

Neuromorphic computing systems that implement artificial neural networks (ANNs) operate differently than conventional computing[44]. ANNs use analog information processing units that collectively interact through interconnected non-linear functions (Supplementary Fig. 1). The fundamental building block of an ANN is a *perceptron*[51], which consists of a linear combination of weighted analog input signals (Supplementary Fig. 2). The analog computation result serves as an input to an activation function that computes the perceptron's non-linear output behavior with soft and hard classifications. The soft classification is observed by using a sigmoid activation function $z = \frac{e^y}{1+e^y}$; where $y$ is a linear signal and serves as an input to the activation function, and the output $z$ is a non-linear analog signal between 0 and 1. The hard classification is observed by using a step activation function; where the output $z$ is a discrete signal receiving only 0 and 1. In considering an adaptation of this model to gene regulatory networks inside individual living cells, it is worthwhile to note that biological pathways often operate in a non-linear fashion and exhibit logarithmic and power law input–output relations, where outcomes are dictated by relative fold-changes rather than absolute levels[28,52,53]. Therefore, to implement ANNs in living cells, we define a log-based version of a perceptron, termed the *perceptgene* (Fig. 1a), whose logarithmic input–output operation makes it more suitable for the non-linear nature of biochemical reactions and gene regulation. The perceptgene implements a *logarithmic classifier* that asymptotically partitions all input values into two classes of output data points (Fig. 1a, right).

In the perceptgene design operating in the logarithmic domain, the perceptron's linear operations of scalar multiplication and summation are transformed to exponentiation (power-law) and multiplication, respectively (Fig. 1a). The perceptgene's sigmoid activation function is described by Michaelis–Menten kinetics at steady state[28,37] ($z = \frac{e^{\ln(y)}}{1+e^{\ln(y)}}$, where $y$ is a scaled protein concentration) and also operates in the logarithmic domain. In order to make the perceptgene operation more compatible with the realities of gene expression, we added basal expression $\beta$ to the activation function ($z = \frac{e^{\ln(y)}+\beta}{1+e^{\ln(y)}+\beta}, \beta \ll 1$) (Fig. 1a). As we discuss below, the perceptron's high-level operations, e.g., weighted multi-input functions, classification, and gradient descent for learning algorithms[46,48] are supported by the perceptgene's log-based computing. Further discussion on perceptron and perceptgene models is provided in Supplementary Notes, Perceptual computing models (Sensitivity, noise and non-linearity analysis).

The perceptgene output is computed as a linear combination in the logarithmic domain (i.e., multiplication) of the weighted inputs and the bias in comparison to the threshold of the activation function ("Methods," Perceptgene abstract model). In the perceptgene implementation, the weights are mainly determined by the Hill coefficients and design topology (e.g., feedback loops). The bias is set by the ratio between the maximum protein (transcription factor) level and the binding affinities of protein-protein/protein-DNA reactions. The fold change of perceptgene output is set by the basal level, which in turn, determines the threshold of the activation function. Practically, we demonstrate our ability to fine-tune perceptgene biological parameters, including Hill coefficients, using well-known strategies for modifying gene regulatory networks (Supplementary Notes, Design principles of neuromorphic gene circuits).

To implement the perceptgene in living cells, we first created a synthetic gene circuit that combines power-law and multiplication functions (Fig. 1b). The power-law function encodes weighted inputs by assigning for each input a particular weight, and the multiplication function aggregates the analog values of the weighted inputs. Our circuit inputs are small molecule inducers: isopropyl β-D-1-thiogalactopyranoside (IPTG) and anhydrotetracycline (aTc), which bind LacI and TetR repressors, respectively. The repressors regulate their own production with auto-negative feedback loops via the $P_{lacO}$ and $P_{tetO}$ promoters[28]. These auto-negative feedback loops implement the input's power-law functions and increase the dynamic range[28]. To implement the multiplication function, we connected combinatorial promoter ($P_{lacO/tetO}$)[54] encoding LacI and TetR operators to the auto-negative feedback loops. The IPTG/aTc regulation of $P_{lacO/tetO}$ promoter via constitutively expressed LacI and TetR implements a conventional Boolean AND logic gate[54], but the regulatory topology described here, auto-negative feedback, converts the promoter's operation into a logarithmically classifier (Eq. 2.13, Supplementary Notes).

Experimentally, the IPTG/aTc transfer function has an input dynamic range of two orders of magnitude for each input (Fig. 1c). Our minimized biochemical model reveals that the power-law coefficients are determined mainly by Hill coefficients describing inducers binding to transcription factors and repressors binding to promoters (Eq. 2.17, Supplementary Notes). Motivated by this analysis, we modified one of the auto-negative feedback loops by replacing $P_{lacO}$ (which has two LacI binding sites) with $P_{lacO1}$ (which has only one LacI binding site) (Supplementary Fig. 17b). The reduced cooperativity of $P_{lacO1}$ resulted in a measured 50% increase in IPTG's power-law coefficient, i.e., its input weight (Fig. 1d). For a negative feedback loop, our mathematical model shows that the IPTG Hill coefficient is inversely proportional to the number of repressor binding sites in $P_{lacO}$ and $P_{lacO1}$ (Eq. 2.20.9, Supplementary Notes). Experimentally, the output of the modified power law and multiplication circuit (using $P_{lacO1/tetO}$) exhibits a three-dimensional linear plot in log-scale coordinations (Fig. 1d) with a NonLinearity degree of 1.33 (Table 1). The NonLinearity degree of gene circuit is computed as the ratio between two slopes – the slope of the input-output transfer function within the linear range, and the slope of a linear curve with the same input dynamic range and maximum fold change of the gene circuit (Supplementary Notes, NonLinearity degree). The kinetics of this optimized power law and multiplication circuit remained stable over the course of approximately ten hours (Supplementary Figs. 92 and 93).

Next, to implement the full perceptgene, we connected the power-law and multiplication function to a customizable activation function. Specifically, we selected $P_{BAD}$/AraC activation, and hence encoded AraC downstream of $P_{lacO/tetO}$, which in turn regulates perceptgene output via promoter $P_{BAD}$ and Arabinose inducer (Fig. 1g). A perceptgene with $P_{BAD}$/AraC activation can be readily customized to perform different computational tasks (e.g., minimum, maximum, and average) by modifying mainly Arabinose concentration (Supplementary Fig. 22b). Arabinose levels tune the $P_{BAD}$/AraC Hill coefficient by converting the transcriptional repression function of free AraC to transcriptional activation of the AraC/Arabinose complex[55]. Experimentally, with low Arabinose levels, the circuit converts an analog pattern of two inputs (IPTG and aTc) into a non-linear function, with a

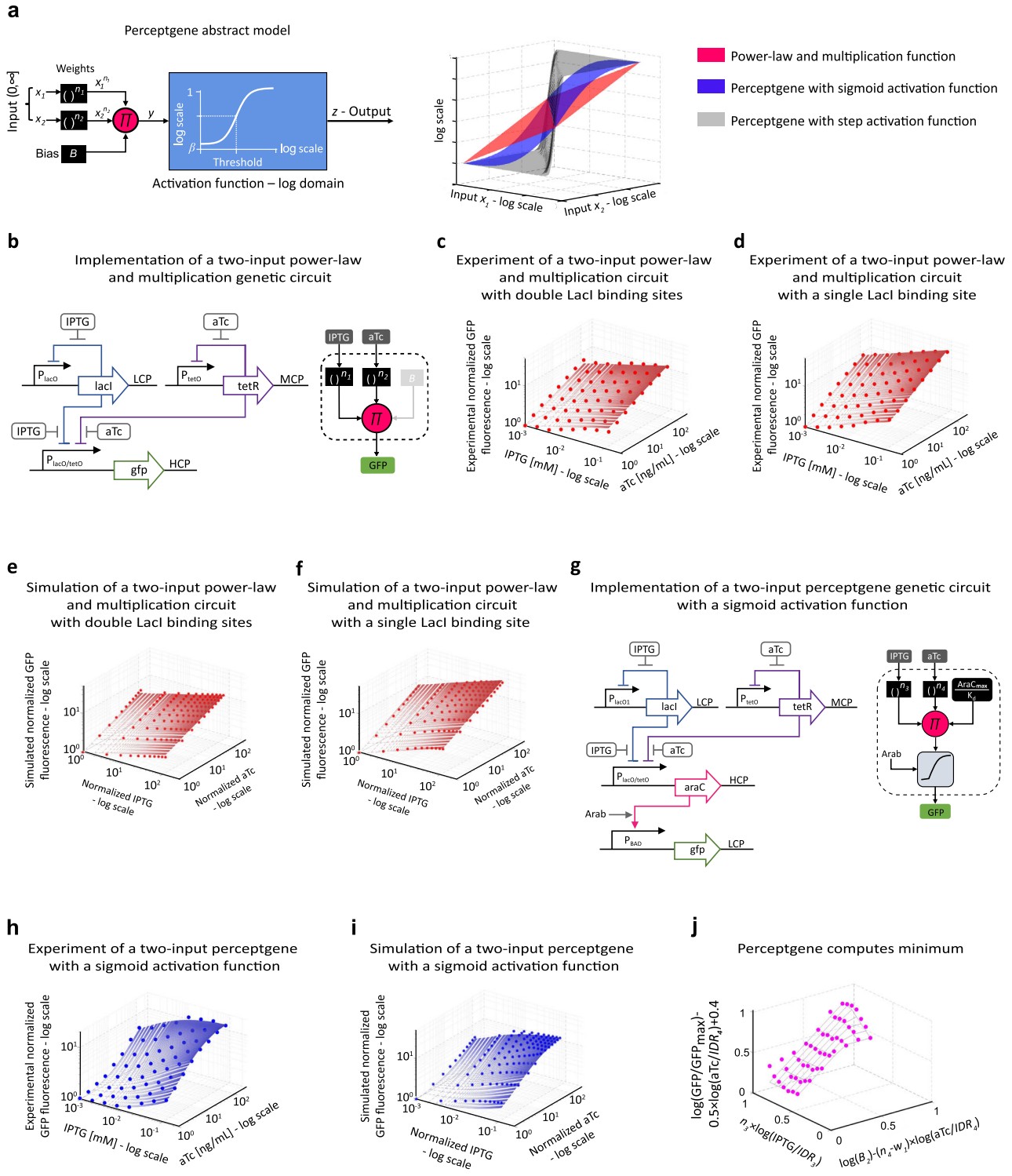

**a** Perceptgene abstract model

**b** Implementation of a two-input power-law and multiplication genetic circuit

**c** Experiment of a two-input power-law and multiplication circuit with double LacI binding sites

**d** Experiment of a two-input power-law and multiplication circuit with a single LacI binding site

**e** Simulation of a two-input power-law and multiplication circuit with double LacI binding sites

**f** Simulation of a two-input power-law and multiplication circuit with a single LacI binding site

**g** Implementation of a two-input perceptgene genetic circuit with a sigmoid activation function

**h** Experiment of a two-input perceptgene with a sigmoid activation function

**i** Simulation of a two-input perceptgene with a sigmoid activation function

**j** Perceptgene computes minimum

NonLinearity degree of 3.8 (Table 1), that allows performing soft classification (Fig. 1h). Our experimental results correlates well with a detailed biochemical model (Fig. 1i and Supplementary Fig. 24).

We then analyzed the computational capabilities of this perceptgene and closely related variants. By extending existing neural computing analysis of perceptrons[56], we proved that the smooth minimum, maximum, and average functions can theoretically be encoded in the operation of perceptgenes using a log-transformed negative rectifier, log-transformed positive rectifier and log-transformed linear activation functions, respectively (Supplementary Notes, Smooth logical functions).

In terms of the perceptgene activation function, for low Arabinose induction levels, $P_{BAD}$ promoter exhibits a shifted and biased log-transformed negative rectifier (Supplementary Fig. 22d); log(AraC) below a threshold ($u_{01}$) result in promoter expression proportional to log(AraC), while concentrations above this $u_{01}$ threshold result in an asymptotically high output (Supplementary Fig. 32c). Theoretically, the log-transformed negative rectifier used in this design can compute the minimum between the $u_{01}$ threshold and log(AraC) ("Methods," Smooth functions). When the log(AraC) is a linear combination of the weighted IPTG and aTc inputs as indicated by this design, the min{$u_{01}$,log(AraC)} operation can be transformed for a minimum between

**Fig. 1 | Perceptgene theory and implementation. a** The Perceptgene model raises each analog input ($x_i$) to the power of its corresponding weight ($n_i$), then multiplies these power-law products to obtain $y = B \cdot \prod x_i^{n_i}$ where $B$ is a bias that shifts $y$ into the desired range, and finally computes the output using $z = \frac{\beta + e^{\ln(y)}}{1 + \beta + e^{\ln(y)}}$ with $z \in [\beta, 1]$, $\beta$ is a minimum (i.e., basal) level. Depicted on the right are power-law and multiplication function, and perceptgenes with sigmoid and step activation functions. **b** The power-law and multiplication circuit for IPTG and aTc inputs. Combinatorial promoter ($P_{lacO/tetO}$) is encoded on a high-copy-number plasmid (HCP) and is regulated by LacI and TetR repressors. $P_{lacO}$, encoded on a low-copy-number plasmid (LCP), and $P_{tetO}$, encoded on a medium-copy-number plasmid (MCP), are regulated through auto-negative feedback loops by LacI and TetR and induced by IPTG and aTc, respectively. Depicted on the right is a block diagram for the genetic circuit operation. **c** The measured transfer function shows the GFP at steady state. Solid line fits to $\left(\frac{IPTG}{1.25}\right)^{0.3375} \cdot \left(\frac{aTc}{0.7}\right)^{0.26}$ with $R^2 = 0.97$. **d** Measured transfer function for a modified circuit where $P_{lacO}$ within the auto-negative feedback loop was replaced by $P_{lacO1}$. Solid line fits to $\left(\frac{IPTG}{1.25}\right)^{0.3375} \cdot \left(\frac{aTc}{0.7}\right)^{0.4375}$ with $R^2 = 0.99$. **e, f** Computed transfer functions of power-law and multiplication circuits with $P_{lacO}$

and $P_{lacO1}$, respectively. IPTG and aTc levels are normalized by their dissociation constants of IPTG-LacI binding and aTc-TetR binding, respectively. Solid line fits to $\left(\frac{IPTG}{1.25}\right)^{0.3375} \cdot \left(\frac{aTc}{0.7}\right)^{0.26}$ and $\left(\frac{IPTG}{1.25}\right)^{0.3375} \cdot \left(\frac{aTc}{0.7}\right)^{0.437}$, $R^2 > 0.95$ respectively. **g** A perceptgene genetic implementation by adding AraC fused to an ssrA degradation tag regulating the $P_{BAD}$ promoter. Depicted on the right is a block diagram for the genetic circuit operation. The bias is equal to the ratio between maximum AraC expression level and $P_{BAD}$ binding affinity. **h, i** Measured and simulated transfer functions of the perceptgene with Arabinose = 0.04 mM. Solid line fits to perceptgene $\frac{\left[\left(\frac{IPTG}{1.25}\right)^{0.3375} \cdot \left(\frac{aTc}{0.75}\right)^{0.4375}/19\right]^{2.2} + 0.045}{\left[\left(\frac{IPTG}{1.25}\right)^{0.3375} \cdot \left(\frac{aTc}{0.75}\right)^{0.4375}/150\right]^{2.2} + 1}$ with $R^2 > 0.9$. **j** A log-transformed smooth minimum computation. The solid lines indicates the minimum between $n_3 \cdot \log(IPTG/IDR_3)$ and $\log(B_2) - (n_4 - w_1) \cdot \log(aTc/IDR_4)$, where $n_3 = 0.337$, $n_4 = 0.473$, $w_1 = 0.3$, $\log(B_2) = -1$, $const_1 = 0.6$, $\log(IDR_3) = 2.1$, $\log(IDR_4) = 2.1$. The $IDR_3$ and $IDR_4$ are input dynamic ranges of IPTG and aTc, respectively. Measured data are normalized by the minimum level. The data represent the average of three experiments. Source data are available in the Source data file.

## Table 1 | NonLinearity degree of neuromorphic gene circuits

| Data | Circuit | Data fitting | $K_d$ | $m$ | $\beta$ | IR | MFC | $\frac{\log(MFC)}{\log(IR)}$ | Nonlinearity |
|------|---------|--------------|-------|-----|---------|-----|-----|------------------------------|--------------|
| Fig. 1d | Power-law and multi-plication (Fig. 1b) | Supplementary Fig. 23a | 150 | 1 | 0.0001 | 128 | 38 | 0.75 | 1.33 |
| Fig. 1h | Perceptgene (Fig. 1g) | Supplementary Fig. 23b | 19 | 2.2 | 0.045 | 128 | 16.5 | 0.57 | 3.8 |
| Fig. 2b | Power-law and multi-plication (Fig. 2a) | Supplementary Fig. 29b | 250 | 1 | 0.0001 | 45 | 40 | 0.96 | 1.03 |
| Fig. 2e | Perceptgene (Fig. 2d) | Supplementary Fig. 31 | 1400 | 1.27 | 0.00045 | 48 | 11.5 | 0.64 | 2 |
| Fig. 2h | Perceptgene (Fig. 2g) | Supplementary Fig. 40b | 400 | 0.95 | 0.0001 | 24 | 16 | 1.06 | 1.08 |

IR is defined as the input range, and MFC is defined as the maximum fold change. To evaluate the Nonlinearity of a neuromorphic gene circuit, we first fit its output to a perceptgene model using $\frac{\left[\left(\frac{AHL}{K_1}\right)^{n_1} \cdot \left(\frac{aTc}{K_2}\right)^{n_2}/K_d\right]^m + \beta}{\left[\left(\frac{AHL}{K_1}\right)^{n_1} \cdot \left(\frac{aTc}{K_2}\right)^{n_2}/K_d\right]^m + 1}$ and extracting Hill coefficient ($m$). Second, we estimate the IR and MFC values. The IR of a perceptgene circuit is equal to the range where the circuit can compute the multiplication of the weighted inputs. For a perceptgene with more than one input, the IR used in Table 1 is calculated as the average of the IRs for each input. The MFC is calculated by the ratio between the maximum and minimum levels of the measured perceptgene output. Finally, the Nonlinearity degree of neuromorphic gene circuits is calculated by $\frac{m}{\log(MFC)/\log(IR)}$. The table above shows that the Nonlinearity increased by more than threefolds when a sigmoid activation function was added (comparing Fig. 1b versus Fig. 1g) and increased by twofolds when a positive rectifier activation function was added (comparing Fig. 2a versus Fig. 2d). The Nonlinearity for perceptgene with a linear activation function (Fig. 2h) equals near one.

IPTG and aTc inputs with an offset proportional to aTc ("Methods," Smooth functions). After graphing the two inputs and the preceptgene output at the logarithmic scale, where the inputs are normalized by their respective dynamic ranges, and the output is normalized by the offset above, smooth minimum computation is revealed, with a standard error of 10% (Fig. 1j, Supplementary Fig. 34, and Supplementary Table 13).

Next, we implemented a perceptgene with an activation function encoding a shifted and biased log-transformed positive rectifier using a modification of the $P_{BAD}$/AraC system (Supplementary Fig. 22d). In particular, $P_{BAD}$ implements a positive rectifier for high Arabinose concentrations and low AraC levels (Supplementary Fig. 32c). For this positive rectifier, $\log(AraC)$ below a $u_{02}$ threshold result in a low constant $P_{BAD}$ activity, while $\log(AraC)$ concentrations above this value elicit promoter expression proportional to $\log(AraC)$. Here, we control AraC level with the Lux and Tet systems (Fig. 2a–d), since we wanted to expand the set of regulatory elements that could be incorporated into multi-perceptgene networks. The inputs to this circuit are aTc and acyl homoserine lactone (AHL). aTc induces expression as above, while AHL binds transcriptional activator LuxR, forms AHL-LuxR complex, and activates promoter $P_{lux}$ (Fig. 2a).

We focused on finding the best AHL/aTc regulation that matches the input dynamic range of $P_{BAD}$/AraC for implementing the positive rectifier function. For AraC expression, we incorporated $P_{lux/tetO}$, a

combinatorial promoter with LuxR and TetR operators[54]. For TetR/aTc regulation, we used the same topology as above. To obtain a LuxR/AHL power-law response, we built a graded auto-positive feedback loop[28] using a weak $P_{lux}$ mutant that broadens the input dynamic range (Supplementary Fig. 26). Similar to the analysis of auto-negative feedback, our biochemical model of auto-positive feedback revealed that the power-law coefficients are determined by the Hill coefficient and binding affinity of LuxR to promoter $P_{lux}$ (Eq. 2.37, Supplementary Notes). We built a library comprising seven different $P_{lux}$ mutant by introducing random mutations to the LuxR operator, which alter LuxR binding affinity to $P_{lux}$ (Supplementary Figs. 27 and 87). The mutant $P_{luxTGT}$ achieved the best match with the input dynamic range of $P_{BAD}$/AraC under high Arabinose levels. Experimentally, the measured AHL/aTc transfer function of $P_{lux/tetO}$ exhibits a power-law and multiplication output response with an input dynamic range for both inputs roughly of two orders of magnitude (Fig. 2b). This transfer function correlated well with our detailed biochemical model (Fig. 2c). To create the full perceptgene, we encoded AraC activator under $P_{lux/tetO}$, which in turn regulated the $P_{BAD}$ promoter (Fig. 2d). Experimentally, with high Arabinose levels, the perceptgene converts an analog pattern of two inputs (aTc and AHL) into a nonlinear function (Fig. 2e) with a NonLinearity degree of 2 (Table 1).

Similar to the analysis of the minimum computation, we observed that a two-input perceptgene with an activation function of a log-

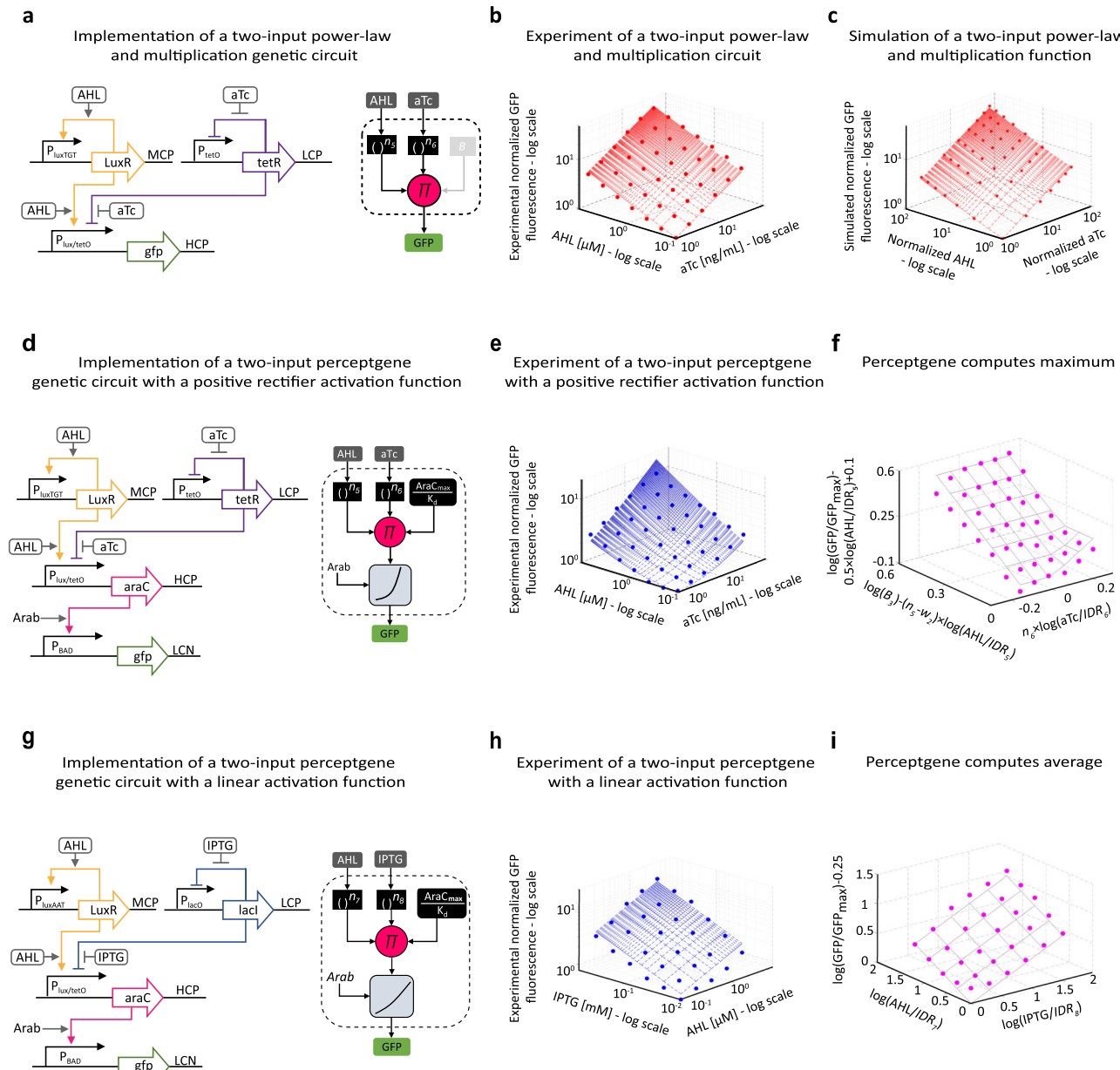

**Fig. 2 | Perceptgenes based on auto-negative and auto-positive feedback loops.**
**a** The power-law and multiplication function circuit for inputs AHL and aTc. Mutant $P_{luxTGT}$ promoter, encoded on an MCP, is regulated by LuxR activator through an auto-positive feedback loop and is induced by AHL. The $P_{tetO}$ promoter, encoded on an LCP, is regulated by TetR through an auto-negative feedback loop and is induced by aTc. A combinatorial promoter $P_{lux/tetO}$, encoded on an HCP, is regulated by LuxR and TetR with output mCherry. Depicted on the right is a block diagram for the genetic circuit operation. **b** Measured AHL, aTc transfer function. Solid line fits to function $\left(\frac{AHL}{1.25}\right)^{0.45} \cdot \left(\frac{aTc}{1.25}\right)^{0.55}$ with $R^2 = 0.95$. **c** Simulated AHL/aTc transfer function based on a detailed biochemical model. The IPTG and AHL are normalized by their dissociation constants of IPTG-LacI binding and AHL-LuxR binding, respectively. Solid line fits to the power-law and multiplication function $\left(\frac{AHL}{1.25}\right)^{0.45} \cdot \left(\frac{aTc}{1.25}\right)^{0.55}$ with $R^2 = 0.96$. **d** Perceptgene with $P_{BAD}$/AraC activation. An ssrA degradation tag was added to AraC. **e** Measured transfer function of the perceptgene circuit with AHL and aTc analog inputs and Arabinose = 0.5 mM. Solid line fits to perceptgene model $\frac{[(\frac{AHL}{1.25})^{0.45}(\frac{aTc}{1.25})^{0.55}/1400]^{1.27} + 0.00045}{[(\frac{AHL}{1.25})^{0.45} \cdot (\frac{aTc}{1.25})^{0.55}/1400]^{1.27} + 1}$ with $R^2 = 0.85$. **f** A log-transformed and normalized smooth maximum computation perceptgene circuit

derived from the experimental results. The solid line indicates the maximum between $n_6 \cdot \log(aTc/IDR_6)$ and $\log(B_3) - (n_5 - w_2) \cdot \log(AHL/IDR_5)$, where $n_5 = 0.55$, $n_6 = 0.45$, $w_2 = 0.22$, $\log(B_3) = -0.2$, $const_2 = 0.1$, $\log(IDR_5) = 1.5$, $\log(IDR_6) = 1.8$. The $IDR_5$ and $IDR_6$ are the input dynamic ranges of AHL and aTc, respectively. **g** A perceptgene for computing a log-transformed average function of AHL and IPTG inputs. The regulatory elements are previously described, except for mutant $P_{luxAAT}$ promoter, which is encoded on an MCP and regulated by LuxR through an auto-positive feedback loop. **h** Measured transfer function of the perceptgene circuit with AHL and IPTG inputs. Solid line fits to perceptgene model $\frac{[(\frac{AHL}{6})^{0.45} \cdot (\frac{IPTG}{7.2})^{0.42}/400]^{0.95} + 0.0001}{[(\frac{AHL}{6})^{0.45} \cdot (\frac{IPTG}{7.2})^{0.42}/400]^{0.95} + 1}$ with $R^2 = 0.85$, and Arabinose = 0.5 mM. **i** A log-transformed and normalized smooth average computation perceptgene circuit derived from the experimental results. Solid line indicates the average operation between $\log(AHL/IDR_7)$ and $\log(IPTG/IDR_8)$ where $\log(IDR_7) = 1.2$, $\log(IDR_8) = 1.5$. The $IDR_7$ and $IDR_8$ are the input dynamic ranges of AHL and IPTG, respectively. Measured data is normalized by the minimum level. Data represent the average of three experiments. Source data are available in the Source data file.

transformed positive rectifier (Fig. 2d induced with high Arabinose) can compute the maximum between two log-transformed analog numbers that are related to AHL and aTc inputs with a particular offset that is related to AHL only (Supplementary Fig. 36). Our analysis demonstrated that a log-transformed positive rectifier function with an input of log(AraC), which is regulated by a linear combination of the weighted AHL and aTc signals as indicated by our design, can compute the maximum between the two signals themselves with an added offset that is related to one of the signals ("Methods," Smooth functions). Therefore, after graphing the AHL and aTc input signals and perceptgene output at a logarithmic scale, normalizing the input signals to their respective input dynamic ranges, and normalizing the output signal by the offset that is related to AHL only, smooth maximum computation is revealed with a standard error of 23% (Fig. 2f and Supplementary Table 14).

The third classification function that we implemented with a single perceptgene was a log-transformed average of two analog inputs IPTG and AHL offset by a constant bias (Fig. 2g). The IPTG and AHL inputs simultaneously regulate combinatorial promoter $P_{lux/lacO}$[54] via graded auto-negative and auto-positive feedback loops. The average operation can be implemented using a perceptgene with a linear activation function and total weights of 0.5 for both inputs. In this circuit (Fig. 2g), we designed the perceptgene's AraC activation function to be linear over the input dynamic range with a slope that yields a total weights of 0.5 (Supplementary Fig. 38). For the IPTG input, we used $P_{lacO1}$/LacI auto-negative feedback with an input weight of 0.95 (Supplementary Figs. 39 and 40). To ensure that AHL's weight matches the IPTG weight, we created a new mutant lux promoter $P_{luxAAT}$ (Supplementary Fig. 27) and incorporated it into a graded positive feedback system. The resulting AHL input weight is 0.85, and closely matches that of IPTG (Supplementary Fig. 40). We then had to compensate for the high IPTG and AHL input weights by fine-tuning the AraC activation function to exhibit a sufficiently shallow slope in the log–log domain. First, we used very high Arabinose concentrations to obtain a $P_{BAD}$/AraC activation function with a low Hill coefficient (Supplementary Fig. 22). Then, our mathematical analysis revealed that for AraC levels slightly lower than the binding dissociation constant of AraC to $P_{BAD}$, along with high Arabinose concentrations, the activation function's slope is approximately 0.5 (Eq. 3.23, Supplementary Notes). Experimentally, with these AraC levels and a high Arabinose concentration, the circuit indeed calculates the log-transformed average of AHL and IPTG (Fig. 2h, with NonLinearity degree around one (Table 1) offset by a normalized value of −¼ with a standard error of 9% (Fig. 2i, Supplementary Fig. 41, and Supplementary Table 15). As with the power law and multiplication circuit, the average circuit output also remained stable over the course of approximately ten hours (Supplementary Fig. 94). Further analysis validating the smooth minimum, maximum, and average functions is provided in Supplementary Notes, Calculations of parameters for a single perceptgene (Supplementary Table 19). We also quantified the signal-to-noise ratio (SNR) for the three circuits based on single-cell measurements (Supplementary Notes, Noise Analysis in Neuromorphic Circuits). We observed that SNR for the power law and multiplication stage is reduced when replacing auto-negative feedback regulation with auto-positive feedback regulation, and that the addition of the activation function (AraC) tends to coalesce the SNR distributions of all three circuits to roughly the same values (Supplementary Fig. 98).

We then assembled combinations of the above perceptgenes into more complex circuits that implement higher-order functions using principles of deep ANNs, including feedforward networks[44]. We first designed a two-layer perceptron network that implements a three-input soft majority function, whose output is "1" (i.e., larger than half of the maximum fold change) when two or more of its three inputs are "1" (Fig. 3a). Our simplified mathematical analysis showed that when considering input values of 0s and 1s, we can evaluate the design

parameters of this perceptron network using a set of linear equations (Table 2). For this analysis, we use linear-domain perceptron activation functions that are approximated as piecewise linear with three regimes (constant low level when perceptgene input is lower than $\gamma_L$, linear as a function of input, and constant high level when perceptgene input is higher than $\gamma_H$). When the inputs to the first perceptron are both low, its output should be low enough such the second perceptron cannot be activated regardless of its input value (Table 2, row 1 for state [000] and row 2 for state [001]). Similarly, when both inputs to the first perceptron are high, its output should be high enough to activate the second perceptron regardless of its input value (Table 2, row 7 for state [110] and row 8 for state [111]). However, when only one of the first perceptron inputs is high, its output should be insufficient to activate the second perceptron by itself (Table 2, row 3 for state [010] and row 5 for state [100]) but high enough to jointly activate the second perceptron if its input is high (Table 2, row 4 for state [011] and row 6 for state [101]).

The mapping from a perceptron network design to a perceptgene network implementation of the majority function involves transformation to the logarithmic domain (Fig. 3b). The implementation comprises two cascaded perceptgenes and a GFP output (Fig. 3c). The perceptgene of the first layer of the cascade has AHL and IPTG inputs, a topology similar to the perceptgene in Fig. 2g, and T7 RNA polymerase output. The second layer perceptgene inputs are T7 RNA polymerase from the first layer perceptgene output and the majority function's third input (aTc). T7 RNA polymerase is modified to include two amber stop codons, which normally block translation[57]. Expression of amber suppressor tRNA supD, which is regulated by aTc, unblocks T7 RNA polymerase translation and activates T7 promoter (Supplementary Notes, Design of 3-input majority function), which activates the GFP output signal.

Our choices of specific weights for implementing the majority function were guided by the simplified linear-domain analysis in Table 2 and a conversion of this analysis into the log-domain (Supplementary Table 21). These analyses essentially yielded the same constraints (Supplementary Table 20 Vs. 21). We determined that the circuit from Fig. 3c can compute majority even with asymmetric weights for AHL and IPTG (Supplementary Fig. 46e, with error of 11%). For $P_{lacO1}$, the IPTG measured input weight was 0.93 (Supplementary Fig. 49). Through random mutations, we found a mutant lux promoter $P_{luxM56}$ which when incorporated into a graded positive feedback system exhibited AHL input weight of 0.48 (Supplementary Fig. 49) and a sufficiently wide input dynamic range (larger than three orders of magnitude, Supplementary Fig. 88). This resulted in a design constraint for bias $B_1$ such that $\gamma_{L1}\text{-min}(0.45, 0.93) < B_1 < \gamma_{L1}$ (Supplementary Table 20). Empirically, $B_1$ is determined by the ratio between the maximal level of AraC and the binding dissociation constant of AraC to $P_{BAD}$. Hence, to satisfy the above constraint we fused an ssrA degradation tag[58] to AraC, resulting in a decrease in $B_1$ by two orders of magnitude from 0.15 to 0.0025 based on simulation results. For the constraint on the input dynamic range of the fist perceptgene activation function, we find that its level should be less than the combined total weight for the AHL and IPTG inputs ($\gamma_{H1} - \gamma_{L1} < n_1 + n_2$).

For the parameter constraints on the second perceptron, we find that the system has a solution only when the internal weight of T7 RNA polymerase is greater than the weight of $P_{tetO}$/aTc (Supplementary Table 20, $m > n_3$), reflecting the intuition that T7 RNA polymerase represents the accumulated value of two inputs to the majority function. With TetR-based negative feedback regulation, the experimentally measured aTc input weight is 0.7 (Supplementary Fig. 50), while T7 RNA polymerase is a monomer activator with weight of 1. However, as our analysis shows, on the one hand the weight of aTc should be larger than the input dynamic range of the second perceptgene activation function ($n_3 > \gamma_{H2} - \gamma_{L2}$), which is approximately equal to 1 (i.e., $P_T$ promoter has a single binding site for T7 RNA polymerase). On the

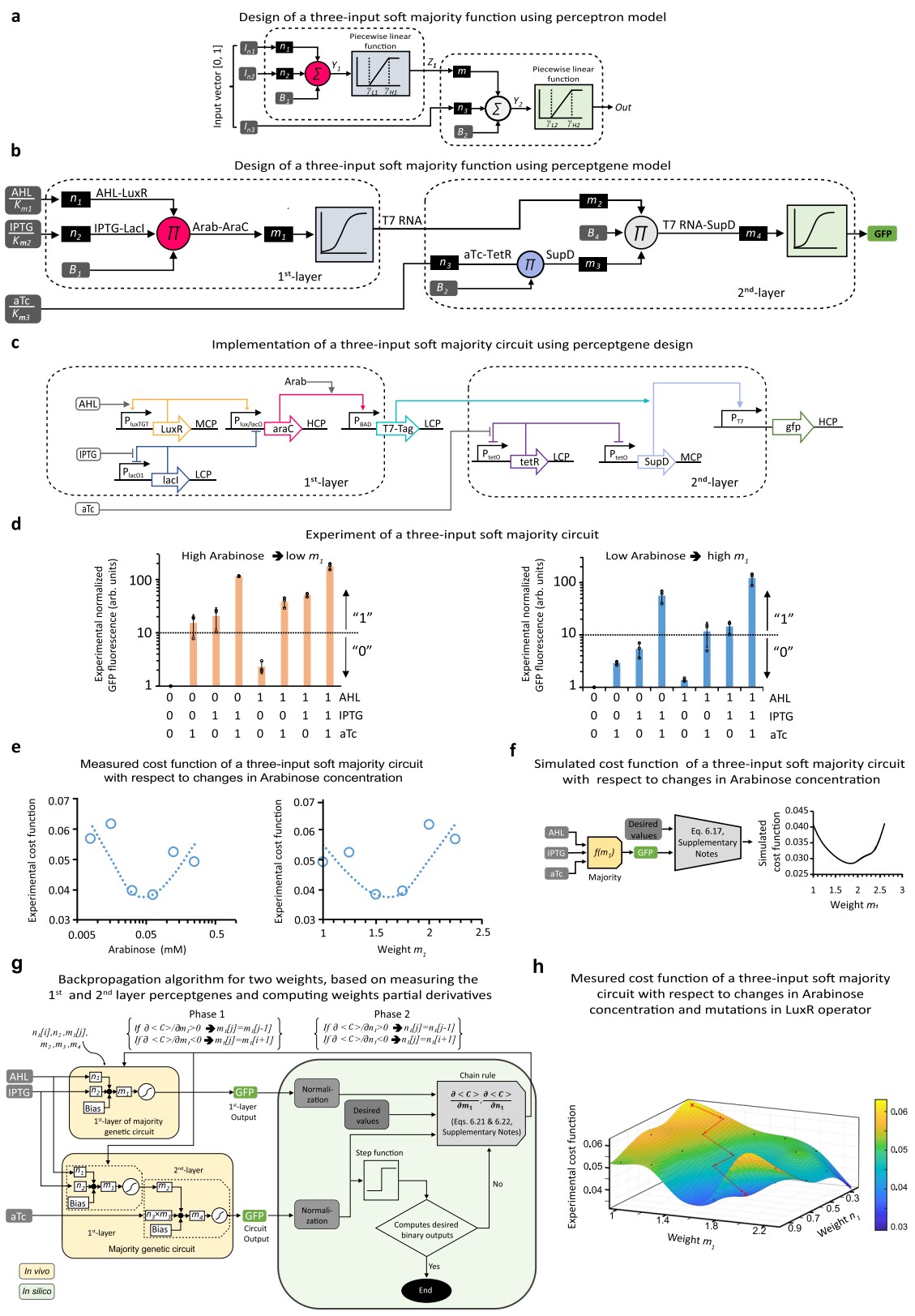

**a** Design of a three-input soft majority function using perceptron model

**b** Design of a three-input soft majority function using perceptgene model

**c** Implementation of a three-input soft majority circuit using perceptgene design

**d** Experiment of a three-input soft majority circuit

High Arabinose ➔ low $m_1$

Low Arabinose ➔ high $m_1$

**e** Measured cost function of a three-input soft majority circuit with respect to changes in Arabinose concentration

**f** Simulated cost function of a three-input soft majority circuit with respect to changes in Arabinose concentration

**g** Backpropagation algorithm for two weights, based on measuring the 1$^{st}$ and 2$^{nd}$ layer perceptgenes and computing weights partial derivatives

**h** Mesured cost function of a three-input soft majority circuit with respect to changes in Arabinose concentration and mutations in LuxR operator

other hand, the weight of aTc should not exceed the weight of T7 RNA polymerase, satisfying the constraint $m > n_3$. Therefore, we incorporated dual SupD binding sites on T7 RNA polymerase[57], which increases aTc input weight by approximately 1.5 (Supplementary Fig. 46) to reach a weight of 1.05. Finally, to ensure that the bias and a single high input of the second perceptgene could not activate its output, we reduced the bias by incorporating a low affinity ribosome binding sequence for T7 RNA polymerase.

After satisfying the various constraint-driven design decisions discussed above by using a high concentration of Arabinose of 0.25 mM that supports near-saturating levels of P$_{BAD}$ activation, the initial version of the soft majority function yielded a system where output for two of the eight AHL/IPTG/aTc input cases were incorrect (left side of Fig. 3d). To address this problem, we focused on improving the performance of the soft majority function by fine-tuning Arabinose, a readily accessible method to alter mainly circuit weight but also

**Fig. 3 | Multilayer perceptgene network and backpropagation algorithm. a** The design of a majority function with two-layer cascaded perceptrons (dashed boxes). **b**, **c** Conversion of the biophysical model of the majority function into a two-layer perceptgene network with inputs AHL, IPTG, and aTc. Network operation is determined by the weights ($m_i$, $n_i$), biases ($B_i$), and activation functions represented by promoters' activity. The first layer inputs are AHL and IPTG, and the output is T7 RNA polymerase, which is regulated by $P_{BAD}$ promoter. The second layer inputs are T7 RNA polymerase and aTc, which their multiplication is achieved via the expression of aTc-regulated SupD and the binding reaction $T7_{RNA} + supD \leftrightarrow T7_{RNA}SupD$. This complex activates the T7 promoter and expresses GFP. The AraC is fused with ssrA degradation tag and encoded on an LCP. The T7 RNA polymerase is regulated by a low binding affinity ribosome-binding sequence. **d** Measured response of the soft majority gene circuit for all eight low/high combinations of the three inputs: AHL [0.1875, 0.3 μM], IPTG [7.8125, 125 μM] and aTc [1.5625, 25 ng/mL]. High Arabinose = 0.25 mM and low Arabinose = 0.03125 mM. The horizontal dashed lines separate between the "0" and "1." **e** The cost function for soft majority circuit under various Arabinose levels and its corresponding weight was estimated using experimental results based on Eq. 6.18, Supplementary Notes. **f** The simulated soft majority circuit cost function was estimated using a logarithmic backpropagation algorithm (Eq. 6.17, Supplementary Notes). **g** Backpropagation algorithm for two weights, based on measuring the first and second layer perceptgenes and then computing partial derivatives for the weights by applying the chain rule using Eqs. 6.21 and 6.22, Supplementary Notes. In phase 1, we update the $P_{BAD}$/AraC weight ($m$) and in phase 2, we update $P_{lux}$/AHL weight ($n_1$). **h** Experimental cost function for two-dimensional weight space. The $P_{BAD}$/AraC weight is regulated by six different pre-selected Arabinose levels (0.25, 0.125, 0.062, 0.031, 0.015M, and 0.007 mM). The weight of $P_{lux}$/AHL is determined by selecting one of a small library of four genetic variants of the LuxR operator (TCTA, GTTG, GAGC, and TGGG). All experimental data represent the average of three experiments. Source data are available in the Source data file.

---

**Table 2 | Truth table of the linear-domain perceptron-based 3-input majority function and evaluation of constraints on the design parameters**

| $I_{n1}$ | $I_{n2}$ | $I_{n3}$ | Out | $Y_1$ | $Z_1$ | $Y_2$ | Constraints on design parameters |
|---|---|---|---|---|---|---|---|
| 0 | 0 | 0 | 0 | Design constraints subsumed by 001 case | | | |
| 0 | 0 | 1 | 0 | $B_1$ | 0 | $B_2+n_3$ | $B_1 < \gamma_{L1}$ <br> $B_2+n_3 < \gamma_{L2}$ |
| 0 | 1 | 0 | 0 | $B_1+n_2$ | $0 \leq f_{A1}(B_1+n_2) < 1$ | $B_2+m \times f_{A1}(B_1+n_2)$ | $B_2+m \times f_{A1}(B_1+n_2) < \gamma_{L2}$ |
| 0 | 1 | 1 | 1 | $B_1+n_2$ | $0 < f_{A1}(B_1+n_2) \leq 1$ | $B_2+n_3+m \times f_{A1}(B_1+n_2)$ | $B_1+n_2 > \gamma_{L1}$ <br> $B_2+n_3+m \times f_{A1}(B_1+n_2) > \gamma_{H2}$ |
| 1 | 0 | 0 | 0 | $B_1+n_1$ | $0 \leq f_{A1}(B_1+n_1) < 1$ | $B_2+m \times f_{A1}(B_1+n_1)$ | $B_2+m \times f_{A1}(B_1+n_1) < \gamma_{L2}$ |
| 1 | 0 | 1 | 1 | $B_1+n_1$ | $0 < f_{A1}(B_1+n_1) \leq 1$ | $B_2+n_3+m \times f_{A1}(B_1+n_1)$ | $B_1+n_1 > \gamma_{L1}$ <br> $B_2+n_3+m \times f_{A1}(B_1+n_1) > \gamma_{H2}$ |
| 1 | 1 | 0 | 1 | $B_1+n_1+n_2$ | 1 | $B_2+m$ | $B_1+n_1+n_2 > \gamma_{H1}$ <br> $B_2+m > \gamma_{22}$ |
| 1 | 1 | 1 | 1 | Design constraints subsumed by 110, 101, 011 cases | | | |

$B_1$ and $B_2$ are the biases of the first layer and second layer perceptgenes, respectively. The three input weights are $n_1$, $n_2$, and $n_3$, while $m$ is the weight of the first layer perceptgene output ($Z_1$) that serves as an input to the second layer perceptgene. $\gamma_{L1}$, $\gamma_{L2}$ and $\gamma_{H1}$ and $\gamma_{H2}$ are the low and high thresholds of the piecewise-linear first and the second activation functions $f_{A1}$ and $f_{A2}$. The $\gamma_{H2}-\gamma_{L2}$, and $\gamma_{H1}-\gamma_{L1}$ are defined as the input dynamic ranges of the activation functions. According to Fig. 3a, the $Y_1$, $Z_1$, $Y_2$ are computed as:

$Y_1 = n_1 \cdot I_{n1} + n_2 \cdot I_{n2} + B_1$

$\begin{cases} Z_1 = 0 \text{ For } Y_1 \leq \gamma_{L1} \\ 0 < Z_1 < 1 \text{ For } \gamma_{L1} < Y_1 < \gamma_{H1} \\ Z_1 = 1 \text{ For } Y_1 \geq \gamma_{L1} \end{cases}$

$Y_2 = n_3 \cdot I_{n3} + m \cdot z + B_2$

$\begin{cases} \text{Output} = 0 \text{ For } Y_2 \leq \gamma_{L2} \\ 0 < \text{Output} < 1 \text{ For } \gamma_{L2} < Y_2 < \gamma_{H2} \\ \text{Output} = 1 \text{ For } Y_2 \geq \gamma_{L2} \end{cases}$

Further analysis of the majority function is provided in Supplementary Table 21.

---

bias, specifically modifying $P_{BAD}$ response (Supplementary Fig. 22b). We measured the performance of the soft majority function under administration of a set of lower Arabinose concentrations. For each individual Arabinose level, we measured the response to eight different AHL/IPTG/aTc combinations (Fig. 3d and Supplementary Fig. 51), which allowed us to compute the overall cost function[59] (i.e,. error) for that $P_{BAD}$/AraC weight (Fig. 3e). Here, the cost function computes the logarithmically mean squared error ($\langle C \rangle = \frac{1}{2N} \sum_{i=1}^{N} \left( \log(\frac{z_{Di}}{z_i}) \right)^2$) (Eq. 6.18, Supplementary Notes) where $N$ is the number of samples (also called the batch size), $z_{Di}$ is the desired output for each state, and $z_i$ is the observed output for these states. The best performance was observed for an Arabinose level between 0.03125 mM and 0.0625 mM that corresponds to an intermediate $P_{BAD}$/AraC weight (Fig. 3e, right side). The experimental results fit the computation of soft majority function. These results correlated well with our computational models of the majority (Supplementary Fig. 47) and cost functions (Fig. 3f, Eqs. 6.16 and 6.17, Supplementary Notes).

Next, we studied whether our circuit's performance may potentially be optimized using a customized backpropagation learning algorithm based on gradient descent[59,60]. Our backpropagation algorithm evaluates how incremental changes in weights for any perceptgene in the network affect overall system performance (i.e., cost function). This evaluation is used iteratively to determine how to update weights in a gradient descent fashion (Fig. 3g). In the first step of the algorithm, the outputs for each of the eight majority function input conditions for both the first layer perceptgene as well as the second layer (i.e., full circuit) are measured experimentally (Supplementary Figs. 57 and 59). The eight output values for each perceptgene are normalized based on the highest level measured and basal expression (Supplementary Table 26). Then, these experimentally derived output values are used in conjunction with a chain rule formula to determine the derivatives of the cost function with respect to changes in the two current weights. These derivatives yield suggestions for the next weights to test (Eqs. 6.21 and 6.22, Supplementary Notes). The algorithm cycles through the weights one at a time by evaluating the sign of the partial derivatives of the cost function ($\frac{\partial C}{\partial n}$) with respect to this particular weight $n$. Based on the sign of the partial derivative, the algorithm updates the weight to the next nearest available value[61]. The algorithm contains two phases; in the first phase, we update $m$ weight, and in the second phase, we update $n_1$ weight. The algorithm repeats this process until either all output values reach their desired binary values or the cost function reaches a local minima.

Using this backpropagation algorithm, we followed the trajectories in a two-dimensional weight space with the $P_{BAD}$/AraC and $P_{lux}$/AHL weights. The $P_{BAD}$/AraC weight is chosen from a set of six different pre-selected Arabinose levels and the weight of $P_{lux}$/AHL is determined by selecting one of a small library of four genetic variants of the LuxR operator (Supplementary Figs. 49 and 56). The four $P_{lux}$ mutations exhibit different weights (0.1, 0.2, 0.27, 0.45) within the operating dynamic range of $P_{lux}$/AHL [0.1875–3 μM]. We exhaustively measured the soft majority function response to eight different AHL/IPTG/aTc combinations across the six different Arabinose concentrations and four $P_{lux}$ mutations, in triplicates (a total of $8 \times 6 \times 4 \times 3 = 576$ samples). This allowed us to pre-compute an overall cost function for each $P_{BAD}$/AraC and $P_{lux}$/AHL weight combination (Fig. 3h). In emulating a backpropagation algorithm, we started at the corner of the weight space with the lowest values of $P_{BAD}$/AraC and $P_{lux}$/AHL weights and iteratively updated these weights. Based on the sign of the weight derivatives, the available higher or lower weight was chosen. The next weight value is either a $P_{lux}$ genetic variant available in our pre-existing library or an Arabinose inducer concentration from an a priori determined set of inducer values. The optimization trajectory culminated in a solution that provides the desired majority function binary output values after three iterations of $P_{lux}$ and Arabinose weight tuning, using information from $8 \times (3 + 3) = 48$ samples. Further experiments and analysis validating the backpropagation algorithm is provided in Supplementary Notes, Gradient descent and backpropagation algorithms in living cells (Supplementary Table 27 and Supplementary Fig. 60).

For our final perceptgene network, we implemented an analog-to-digital converter (ADC; Supplementary Fig. 61), useful for a variety of intracellular and extracellular biosensing applications. The conversion of analog information into digital encoding is a classification problem that is efficiently solved by ANN architectures[62] (Supplementary Notes, Design and implementation of 2-bit log-ADC). We evaluated three designs, starting from a perceptgene adaptation of a classical ADC perceptron network[62], a second design that adds two inhibitory regulatory links, and a third design that improves the fidelity of the digital output signals. In our first design (Fig. 4a), individual perceptgenes convert the analog input into digital bits starting from the most significant bit (MSB) to the least significant bit (LSB). While the analog input signal is routed to all perceptgenes, the computation from the MSB also contributes a negative weight to the LSB. This effectively subtracts the higher order bit from the LSB's analog input signal (Supplementary Figs. 62 and 63). We designed a gene network that uses transcriptional interference[63] to perform subtraction (Supplementary Fig. 65). We first checked whether $P_{lux}$ promoter activity can be subtracted from $P_{BAD}$ promoter activity by arranging the promoters in a convergent orientation[64]. However, experimental analysis of this subtraction method revealed that in addition to observing the desired increase in the input threshold required for output promoter activation, there was also undesirable repression of maximal promoter activation (Supplementary Fig. 65). Such repression can affect ADC performance for high input by displaying (1,0) instead of (1,1). To alleviate repression of maximal promoter activation caused by the transcriptional interference, we introduce a second regulatory element TetR, which represses a hybrid version of the convergent $P_{lux}$ promoter ($P_{lux/tetO}$, Supplementary Fig. 68). In the revised ADC design, TetR is regulated by the MSB to nullify the subtraction of the MSB from the LSB only for high input levels.

The revised ADC gene circuit comprises four main elements: the AHL input stage, the MSB, the MSB subtractor, and the LSB (Fig. 4b, c). For the AHL input, graded positive feedback with LuxR increases the input's dynamic range, as above (Supplementary Fig. 26). To compute the MSB, AHL activates expression from $P_{lux}$ encoded on a medium copy number. Computation of LSB involves AraC activation of $P_{BAD}$ as well as down-regulation by transcriptional interference[63] from convergent promoter $P_{lux/tetO}$, which is oriented in the opposite direction

to $P_{BAD}$. This transcriptional interference implements subtraction of MSB from the LSB for intermediate levels of AHL, but not for high levels of AHL. This resulted in four distinct outputs states for the two-input ADC (green and red lines in Fig. 4d). An experiment using the initial circuit design in which $P_{lux}$ promoter (without tetO operator) replaces transcriptional interference promoter $P_{lux/tetO}$ shows low LSB levels under high AHL concentrations (purple line in Fig. 4d). Further experiments are provided in Supplementary Notes, Synthetic Data converters (Supplementary Figs. 64–70).

To improve the accuracy of the LSB output for the (1,0) state, we revised the ADC design to compute LSB by including two separate perceptgenes ($LSB_{low}$, and $LSB_{high}$) with the same GFP output (Fig. 4e, f). This technique of aggregating circuit output from two promoters has been used successfully in previous synthetic gene circuit designs[19,28]. The $LSB_{low}$ perceptgene responsible for exhibiting high LSB output for the (0,1) state is configured similarly to the LSB perceptgene used in the previous ADC design. One difference is removal of TetR and use of $P_{lux}$ instead of $P_{lux/tetO}$ for transcriptional interference in order to regulate LSB by MSB even for high AHL (Supplementary Figs. 67 and 68). The $LSB_{high}$ perceptgene responsible for exhibiting high LSB output for the (1,1) case consists of promoter $P_{rhl}$ that is directly regulated by AHL-LuxR but requires high levels of AHL for activation (Supplementary Fig. 73). This revised design yielded four distinct digital states with improved performance for the (1,0) state, albeit with inclusion of an undesirable narrow transitional state between (0,1) and (1,0) (Fig. 4g). The revised ADC maintained stable output for approximately 10 h (Supplementary Figs. 95–97). In comparison to previous synthetic data converters[25,65], our logarithmic domain analog-to-digital converter performs a complex computation that encodes the digital output value using multiple bits of information. The circuit in ref. 25 implemented a digitizer, where one single analog input is converted to three discrete outputs, rather than a 2-bit ADC. The circuits in ref. 65 integrate multiple analog inputs and display multiple digitals outputs using logic gate design. In terms of efficiency, the computation in this study requires only two transcription factors.

## Discussion

Emerging applications in synthetic biology benefit from implementations that achieve complex regulatory functions using a minimal number of parts, based on designs that can be optimized with efficient algorithms. The existing paradigms of digital and analog biological circuits often fail in these regards. Instead, here we introduced the notion of genetic circuits that implement logarithmic domain ANNs, with an approach that supports flexible choices of weights and biases for circuit optimization (Supplementary Notes, Design principles of neuromorphic gene circuits). Logarithmic operation, which is based on fold-change regulation rather than absolute changes, is suitable for biological information sensing and transduction in both natural and synthetic systems[28,53,66] (Supplementary Notes, Perceptual computing models).

To gain further insight into the benefits of performing log-domain perceptgene computing, we compared the perceptgene behavior to a linear-domain perceptron that uses a biologically relevant Michaelis–Menten activation function instead of a sigmoid function (Supplementary Fig. 3b). First, with the same input dynamic range for both models, our analysis showed that the perceptgene successfully classified an analog signal into low and high levels, in contrast to the modified perceptron failed (Supplementary Fig. 5). This analysis indicates that logarithmic domain computing is more suitable than linear domain computing for our neuromorphic gene circuits. Second, our theoretical sensitivity analysis also showed that logarithmic domain neuromorphic computing is robust to noise at low signal concentrations, whereas the performance of analog circuits is often poor for such signals (Supplementary Fig. 10). Third, both the perceptgene and perceptron designs support only a limited number of distinct inputs. While

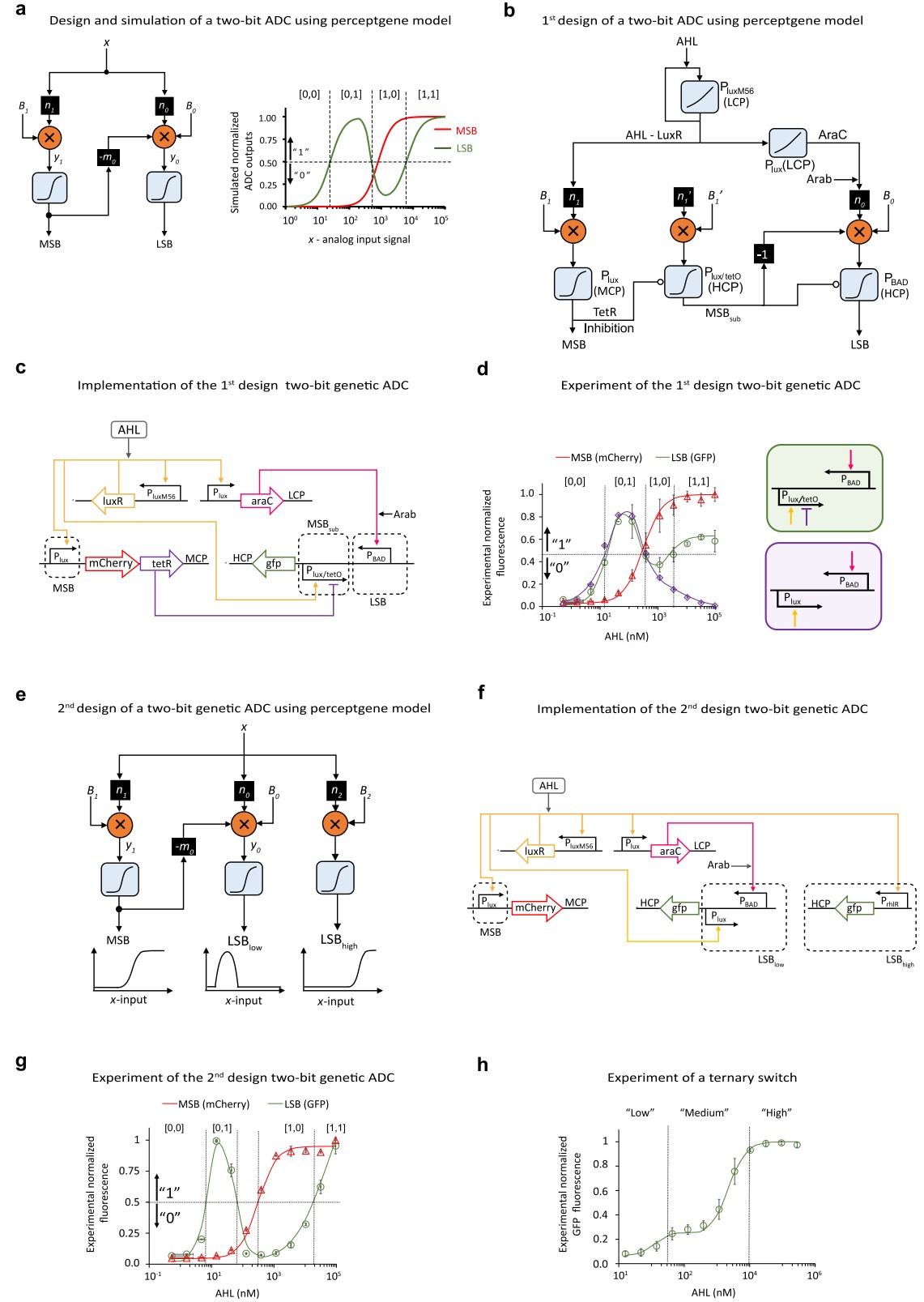

**a** Design and simulation of a two-bit ADC using perceptgene model

**b** 1st design of a two-bit ADC using perceptgene model

**c** Implementation of the 1st design two-bit genetic ADC

**d** Experiment of the 1st design two-bit genetic ADC

**e** 2nd design of a two-bit genetic ADC using perceptgene model

**f** Implementation of the 2nd design two-bit genetic ADC

**g** Experiment of the 2nd design two-bit genetic ADC

**h** Experiment of a ternary switch

the perceptgene is mainly limited by the ability of the biological part structure to integrate multiple analog signals in regulating gene expression, the perceptron is limited by the intrinsic noise generated during signal aggregation (Supplementary Table 1). Given these constraints, implementing a linear-domain perceptron in living cells may nonetheless be practical for applications that focus on a single layer with multiple inputs of highly expressed analog signals, such as cell-free

systems[67] and microbial consortia[68]. The observed transfer functions obtain high sharpness in the case of cell-free systems via substrate saturation and in the case of the microbial consortia via multicellular positive feedback using secreted small molecule inducers that bind transcription factors. However, while a few examples exist, engineering a large library of synthetic biology devices that exhibit sharp responses when operating inside individual cells remains a challenge.

**Fig. 4 | Genetically encoded data converters based on neural network principles. a** Perceptgene-based circuit design and simulation of a two-bit analog-to-digital converter (ADC). The input "$x$" is an analog signal, the outputs are digital bits: most significant bit (MSB) and least significant bit (LSB). **b** The modified perceptgene design of 2-bit ADC based on analyzing gene networks. A subtraction of the MSB from the LSB using transcriptional interference (denoted by "−1") is required. **c** Genetically encoded two-bit ADC. The circuit converts AHL concentration into LSB and MSB. Positive feedback regulation of AHL via mutant $P_{luxMS6}$ promoter linearizes the response. The MSB, encoded on an MCP, is computed by the $P_{lux}$ promoter, which regulates mCherry and TetR. The LSB, encoded on an HCP, receives the linearized AHL, which activates $P_{BAD}$ promoter and expresses GFP. The MSB subtractor (MSB$_{sub}$, encoded on an HCP) regulates $P_{BAD}$ promoter via transcriptional interference ($P_{lux/tetO}$, oriented in the opposite direction to $P_{BAD}$). **d** Measured and simulated response of a 2-bit ADC (Arabinose = 0.4 mM). Red triangles show the measured MSB mCherry. Green circles show the LSB

GFP with $P_{lux/tetO}$ promoter, purple diamonds show the GFP with $P_{lux}$ (Arabinose = 0.4 mM). The $P_{lux/tetO}$ and $P_{lux}$ configurations of the convergent promoters are shown below the graph. **e** Design of a 2-bit ADC using three perceptgenes, two of which are used for computing the LSB; LSB$_{low}$ is activated only under low AHL concentrations, and LSB$_{high}$ is activated only under high AHL concentrations. The MSB topology is similar to Fig. 4b. **f** Genetic implementation of the 2-bit ADC from Fig. 4e. In comparison to Fig. 4c, we removed TetR, replaced $P_{lux/tetO}$ with $P_{lux}$ to obtain LSB$_{low}$, and implemented LSB$_{high}$ using $P_{rhl}$ promoter that is activated by high levels of AHL. **g** Measured and simulated response of the 2-bit ADC from Fig. 4f (Arabinose = 0.06 mM). **h** Measured and simulated response of the ternary switch (Arabinose = 0.4 mM). The horizontal dashed lines in Fig. 4d, g separate between the "0" and "1". Computational simulations are depicted by lines. The data represent the average of three experiments, and is denoted by various data point markers and std. dev. Measured data is normalized by the minimum level. Source data are available in the Source data file.

The design principles of logarithmic domain ANNs exhibit collective resilient properties, offer efficient parallel execution of complex functions, require a small number of components to produce required results, and provide scalability for deep networks. These properties enable construction of systems that can be readily adapted to customized functions by supervised optimization algorithms[46,48]. We began to explore these properties experimentally by creating several neuro-inspired genetic circuits that encode minimum, maximum, and average functions, each comprising a single perceptron with analog outputs. The minimum and maximum functions are widely used in fuzzy logic computing to implement conjunction, disjunction, implication, equivalence, and negation[69]. For biological systems, these functions can, for example, be useful for situations that require graded expression levels that are precisely determined by multiple inputs, as opposed to simple ON/OFF expression. The minimum operation may improve safety and efficacy of genetic circuits for cancer immunotherapy because it enables recognition of cancer biomarkers in a manner reminiscent of an AND logic gate[70,71]. But in contrast to AND logic, the minimum operation will activate the immunotherapy at levels proportional to the biomarkers detected, which may reduce undesirable effects such as cytokine storms. The average function is useful for engineering biological systems able to tolerate noise and compensate for distortion of biological signals.

Our neural gene circuits may also provide an alternative framework for building logic functions that reduce usage of cellular resources. For example, the implementation of our three-bit soft majority function requires 15 biological parts (i.e., promoters and genes), in comparison to 22 parts used for the state-of-the-art three-bit majority implementation based on the digital abstraction[2]. To improve the accuracy of our majority circuit and get distinct low/high outputs, we may connect the circuit output with a single recombinase protein that usually has a sharp response (Supplementary Fig. 52). The design of a two-bit ADC in mixed-signal computation required three logic stages[72] and approximately ten biological parts[25], whereas our neuro-inspired two-bit ADC in which the MSB regulates the LSB in an analog manner requires only two transcription factors. In terms of a relevant but more complex computation, theoretical analysis of a 2-bit Full Adder implementation comparing the neuromorphic versus digital approaches is provided in Supplementary Fig. 91. Besides minimizing circuit sizes, our perceptgene networks also operate with low expression levels, mainly in order to maintain low bias levels. For instance, our circuits' proteins are expressed at low levels either via usage of weak ribosome binding sites, weak RNA polymerase binding sites, or by fusion with ssrA degradation tags. In contrast, digital systems often attempt to operate with significant noise margins, and hence high expression levels for ON values.

Another important property of ANNs is the ability to efficiently fine-tune and repurpose their function by changing weights and biases. For example, by increasing the bias and weights of a perceptgene from

low to high, its computation can be modified from AND to OR (Supplementary Fig. 75). This ability is experimentally demonstrated by inducing a small molecule that controls transcription factor sequestration[73] via protein–protein interactions (Fig. 5a). Specifically, we selected ExsD to shunt ExsA from binding to the $P_{exsA}$ promoter, which activates the GFP expression. We first used the IPTG/LacI system to regulate the expression of ExsA and AHL to induce the expression of anti-activator ExsD through regulating $P_{luxTGT}$/LuxR. The experiment of IPTG-GFP transfer functions for various AHL concentrations indicated that this sequestration significantly decreased the input IPTG's Hill coefficient and hence modulated the preceptgene's internal weight (Fig. 5c). Also, the experiment results of IPTG-GFP revealed an increase in the dissociation constant of IPTG and a decrease in the maximum fold change of $P_{exsA}$ (Fig. 5c). However, since the bias is inversely proportional to the dissociation constant ("Methods," Perceptgene abstract model), and the maximum fold change is inversely proportional to the threshold of the perceptgene's activation function, we concluded that titrating AHL affects mainly the perceptgene's internal weight. Then, we built a two-input perceptgene circuit using the combinatorial promoter ($PI_{acO1/tetO}$) and auto-negative feedback loops encoding LacI and TetR, as shown in Fig. 1b. In the new design (Fig. 5d), we connected the power-law and multiplication function to $P_{exsA}$/ExsA activation function by encoding ExsA downstream of $P_{lacO/tetO}$. To control the internal weight, we induced ExsD by AHL using a weak $P_{lux}$ mutant that broadens the AHL dynamic range. We induced the gene circuit with (1) AHL = 0, which led to a high internal weight observing GFP signal with an OR-logic gate manner (Fig. 5e), (2) AHL = 0.34 μM, which led to a low internal weight observing GFP signal with an AND-logic gate manner (Fig. 5f). To improve the accuracy of the OR circuit and get distinct low and high outputs (Fig. 5g), we increased the input weights by enhancing the strength of the auto-negative feedback loops. In the modified design, we constructed the auto-negative feedback loops of $P_{lacO1}$ and $P_{tetO}$ in medium plasmid copy numbers instead of low copy numbers, which resulted in a more than 25% increase in IPTG's and aTc's power-law coefficients (Supplementary Fig. 79). These experimental results are supported by our biochemical models (Supplementary Fig. 21).

As another example, the ADC circuit can be easily reconfigured to function as an AHL-induced ternary switch with distinct low, medium, and high output states (Supplementary Notes, Design and implementation of ternary switch). Analysis of the model suggested this can be achieved by enhancing activation of the LSB perceptgene (Supplementary Fig. 76). Accordingly, we experimentally administered high levels of Arabinose, and observed the desired ternary response to AHL (Fig. 4h). As yet another example, we analyzed the behavior and topology of the majority circuit and experimentally demonstrated that replacement of promoter $P_{tetO}$ with combinatorial promoter $P_{lux/tetO}$ yields a new logic function "AHL OR [IPTG AND aTc]" (Supplementary Fig. 78). Furthermore, the neuro-inspired design of the three-input

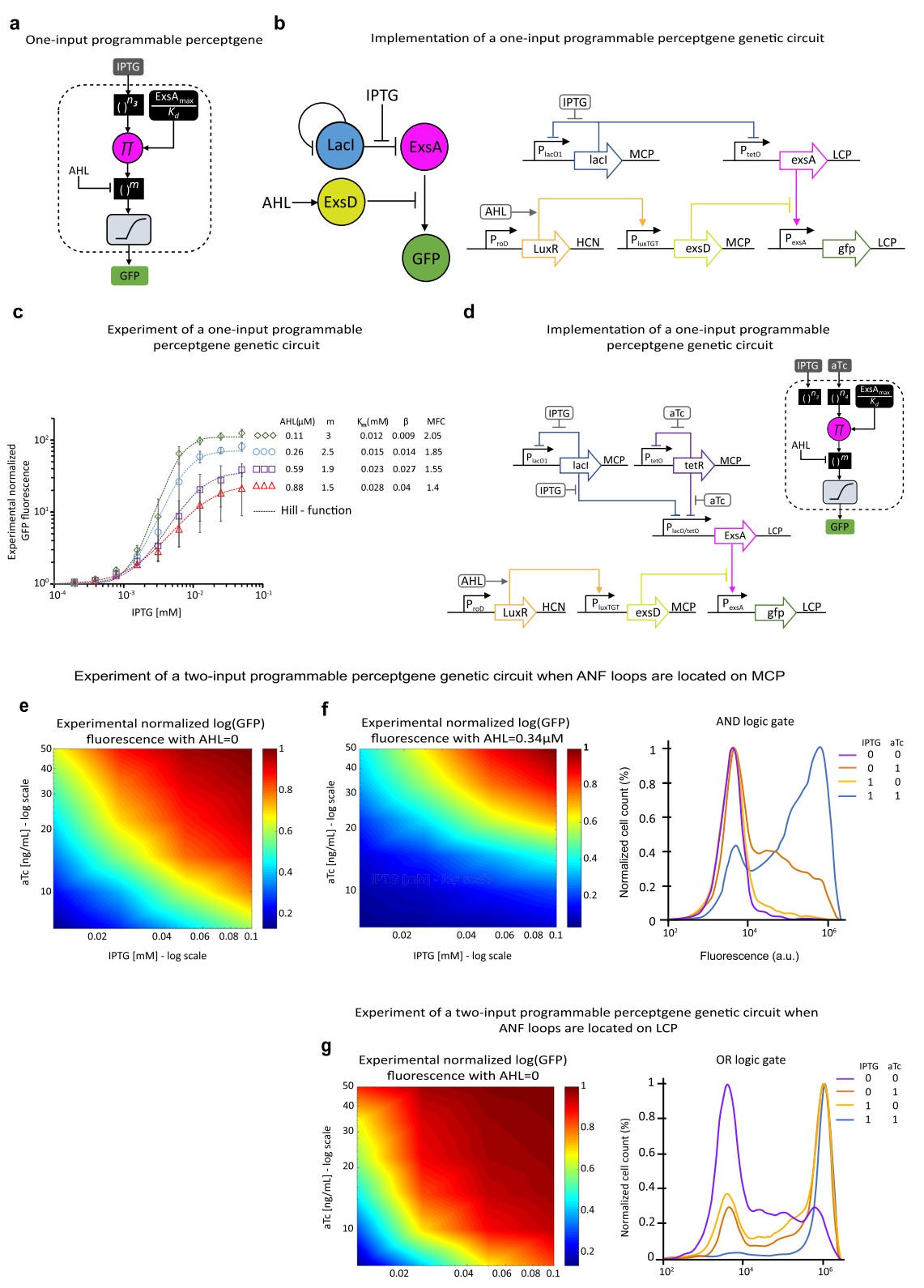

**a** One-input programmable perceptgene

**b** Implementation of a one-input programmable perceptgene genetic circuit

**c** Experiment of a one-input programmable perceptgene genetic circuit

| AHL(µM) | m | $K_m$(mM) | β | MFC |
|---|---|---|---|---|
| 0.11 | 3 | 0.012 | 0.009 | 2.05 |
| 0.26 | 2.5 | 0.015 | 0.014 | 1.85 |
| 0.59 | 1.9 | 0.023 | 0.027 | 1.55 |
| 0.88 | 1.5 | 0.028 | 0.04 | 1.4 |

Hill - function

**d** Implementation of a one-input programmable perceptgene genetic circuit

Experiment of a two-input programmable perceptgene genetic circuit when ANF loops are located on MCP

**e** Experimental normalized log(GFP) fluorescence with AHL=0

**f** Experimental normalized log(GFP) fluorescence with AHL=0.34µM

AND logic gate

Experiment of a two-input programmable perceptgene genetic circuit when ANF loops are located on LCP

**g** Experimental normalized log(GFP) fluorescence with AHL=0

OR logic gate

majority circuit allowed us to optimize the error by changing the weights of $P_{BAD}$/AraC and $P_{lux}$/AHL in similar manner to back-propagation algorithm. Supplementary Notes, Design principles of neuromorphic gene circuits, provides experimental demonstration of modulating weights and biases using a variety of additional biological mechanisms including transcription factor sequestration, steric hinderance, and operator sequences.

Our theoretical and experimental study shows that perceptgene networks can utilize a broad range of biological regulatory mechanisms and accordingly, we expect to be able to implement these networks with other modalities of biological regulation (e.g., protein–protein interactions[74,75] and RNA devices[76,77]) and in different organisms. Flexibility in implementation approaches will help employ such neural networks to address a wide range of industrial, diagnostic, and

**Fig. 5 | A programmable two-input perceptgene network. a** A block diagram for a programmable perceptgene with a single input (IPTG). The AHL inducer controls the internal weight (*m*) of the perceptgene output. **b** High-level genetic circuit diagram and implementation. The design is based on protein sequestration where ExsD shunts ExsA from activating GFP expression encoded on HCP. The IPTG and LacI, encoded on MCP, interact via an auto-negative feedback loop to regulate ExsA that binds P$_{exsA}$. In our implementation, we used P$_{lacO1}$ promoter to implement the auto-negative feedback, and a combinatorial promoter (P$_{lacO/tetO}$) to express ExsA encoded on LCP. In the absence of TetR, the combinatorial promoter activity is determined only by LacI and IPTG. The LuxR/AHL complex regulates the expression of ExsD. LuxR encoded on HCP is expressed by a constitutive promoter (P$_{roD}$). To control ExsD encoded on MCP, we used a mutant P$_{lux}$ promoter (P$_{LuxTGT}$). **c** Measured IPTG transfer functions under different AHL conditions. The dotted lines are fittings to Hill-functions normalized by their minimum levels $\frac{1}{\beta} \frac{\left(\frac{AHL}{K_{eff}}\right)^{m_{eff}} + \beta}{1 + \left(\frac{AHL}{K_{eff}}\right)^{m_{eff}}}$. MFC is the maximum fold change: MFC = log(1/$\beta$) ("Methods," Perceptgene

abstract model). **d** A programmable perceptgene with IPTG and aTc analog inputs computes logic operations. The power-law and multiplication circuit with inputs IPTG and aTc from Fig. 1b was connected with the P$_{exsA}$/ExsA system. The ExsD is controlled by AHL and the auto-negative feedback loops for LacI and TetR are encoded on MCP. **e** Measured IPTG/aTc transfer function for AHL = 0. **f** Measured IPTG/aTc transfer function for AHL = 0.34 μM yields AND-logic gate manner. Depicted distributions on the bottom are the experimental flow cytometry data for the four low/high combinations of the two inputs IPTG [0.125, 0.1 mM] and aTc [6.5, 50 ng/mL]. **g** Modified circuit: the auto-negative feedback loops for LacI and TetR are encoded on LCP. Measured IPTG/aTc transfer function for AHL = 0 yields an OR-logic gate manner. Depicted distributions on the bottom are the experimental flow cytometry data for the four low/high combinations of the two inputs IPTG [0.125, 0.1 mM] and aTc [6.5, 50 ng/mL]. Measured data is normalized by the minimum level. Source data are available in the Source data file.

therapeutic applications (Supplementary Notes, Potential Applications of Synthetic Neuromorphic Circuits). For example, bioengineers could use analog-to-digital converters for selecting which specific combinations of several genes to express based on administration of a single inducer. Also, with a single inducer, bioengineers could harness the ternary switch to select between one of multiple expression levels for a given gene in a robust and noise resistant fashion. These circuits could also form the basis for engineering more sophisticated cellular bio-sensing, e.g. multi-bit classifiers with higher precision than their digital counterparts and increased robustness relative to their analog counterparts. These regulation and sensing capabilities are valuable both in biomanufacturing and therapeutic contexts.

We anticipate that the framework described here can efficiently optimize gene circuit design by applying data-driven algorithms, which determine how one can modify the circuit's design parameters by measuring the circuit's output signals in each step during the optimization process. Future efforts will include codifying the design principles of neuromorphic gene networks such as development of effective mechanisms to combine coarse-grain and fine-grain control over weights and biases (Supplementary Notes, Design principles of neuromorphic gene circuits). ANNs are also compatible with the digital and analog computing platforms. One can leverage the specific advantages of each of these three platforms in a synergistic fashion to create an efficient, accurate, and scalable hybrid approach for robust genetic engineering of living cells (Supplementary Fig. 99). Future efforts will also focus on developing computer-aided design tools that combine ANN design principles with linear equations and logic gates (e.g., Cello[2]) to automate biological engineering. Analogous to the manner by which natural signaling pathways combine a variety of regulatory modalities, we anticipate that future synthetic gene networks integrate the different approaches mentioned above in ways that are particularly suitable for the biological substrate.

## Methods
### Perceptgene abstract model
Supplementary Figs. 42 and 43 describe a general structure of a multi-input perceptgene. The collective signal is regulated by a combinatorial (hybrid) promoter (P$_{Hybrid}$), which includes multiple DNA regulatory binding sites (where either activator or repressor can bind and regulate the promoter). The activity of the combinatorial (hybrid) promoter can be expressed as[78]:

$$P_{Hybrid} = \prod_{i=1}^{N} F_i \left( \left( \frac{Input_i}{K_{mi}} \right)^{h_i} \right) \quad (1)$$

*N* is the number of the inputs, $k_{mi}$ is the dissociation constant of binding Input$_i$ to the appropriate DNA site within the hybrid promoter,

and $h_i$ is the Hill coefficient. $F_i$ is a regulation factor ($0 \leq F_i \leq 1$) that describes the binding of the inputs to DNA sites and usually has a nonlinear response with respect to the inputs. The nonlinearity arises from intermolecular interactions and network topologies of biochemical reactions. Applying negative and positive feedback loops can broaden the input dynamic range of $F_i$. The promoter activity within the desired input dynamic range is:

$$P_{Hybrid} = \prod_{i=1}^{N} \left( \frac{Input_i}{K_i} \right)^{n_i} \quad (2)$$

$n_i$ is determined by $h_i$ and circuit topologies, and $k_i$ is a normalization constant that depends on $k_{mi}$ and circuit topologies (Eqs. 2.17 and 2.37, Supplementary Notes). A similar form to Eq. 2, "Methods," can be observed for biochemical binding reactions to build a complex ($E + S \leftrightarrow ES$). The Collective signal (transcription factor level) is described by an ordinary differential equation that models the production rate ($\alpha$) multiplied by the hybrid promoter activity and subtracted by the degradation process represented by protein half-life or cell growth rate ($\tau$):

$$\frac{dCollective}{dt} = \alpha \cdot P_{Hybrid} - \frac{Collective}{\tau} \quad (3)$$

In the steady state (dCollective/d*t* = 0), and by substituting Eq. 2, "Methods" into Eq. 3, "Methods," we obtain:

$$Collective = \prod_{i=1}^{N} \alpha \cdot \tau \cdot \left( \frac{Input_i}{K_i} \right)^{n_i} \quad (4)$$

where $\alpha \cdot \tau$ has units of concentration, and it equals the maximum level of produced Collective signal (e.g., protein, Eq. 2.13, Supplementary Notes). When the Collective signal binds to the output promoter, its activity is initiated:

$$P_{Out} = \frac{\left( \frac{Collective}{K_d} \right)^m + \beta}{1 + \left( \frac{Collective}{K_d} \right)^m} \quad (5)$$

where $\beta$ is the basal level of the promoter, $K_d$ is the dissociation constant of binding the Collective signal to P$_{out}$ promoter, and *m* is the Hill coefficient. Equations 4 and 5, "Methods," can yield:

$$\begin{cases} y = \left( \prod_{i=1}^{N} B \cdot x_i^{n_i} \right)^m \\ P_{Out} = \frac{y + \beta}{1 + y} \end{cases} \quad (6)$$

$x_i = \text{Input}_i/K_i$ is a normalized input, and $B = \alpha \cdot \tau / K_d$. The model in Eq. 6, "Methods," includes three design parameters:

(1) Network weights ($n_i$ and $m$): are represented by the effective Hill coefficients, and calculated by the log domain slope of the regulated promoter's dosage response curve. The Hill coefficients depend on the biological cooperativity of proteins, the number of binding sites in the promoter, the protein quaternary structure (the number of subunits that interact with each other and arrange themselves to form a final protein), and the design topology.

(2) Bias ($B$): The bias is determined by the ratio between the maximum expression level of a transcription factor (Collective signal) and its binding dissociation constant to DNA. The maximum protein or transcription factor expression level of the Collective signal in Eq. 4, "Methods" is equal to the product of the protein production rate determined by the translation/transcription rates, mRNA/protein half-lives, and cell growth rate. The $K_d$ dissociation constant is determined by binding affinities in protein–protein or protein–DNA reactions.

(3) Activation function depends on promoter activity and is given by the Michaelis–Menten model. The basal level has two significant roles in determining the behavior of the perceptgene model. First, it preserves the maximum fold change ($MFC$) (i.e., output dynamic range) in the logarithmic scale: $MFC = \log(1/\beta)$, and second, it sets the effective threshold (Th) of the perceptgene. ($10^{-\log(1/\beta)/2} = \frac{\text{Th}+\beta}{1+\text{Th}+\beta}$, Supplementary Notes, Calculations of parameters for a single perceptgene).

## Smooth functions

Starting with smooth minimum function: When $P_{\text{BAD}}$ promoter is induced with low Arabinose levels, we can approximate its activity within the desired input dynamic range as a shifted and biased log-transformed negative rectifier activation function (Supplementary Fig. 32c):

$$P_{\text{BAD}} \propto \begin{cases} \log(\text{AraC}) & \log(\text{AraC}) < u_{01} \\ \text{constant} & \log(\text{AraC}) > u_{01} \end{cases} \quad (7)$$

Numerically, Eq. 7, "Methods," can be approximated as (Supplementary Notes, Smooth logical functions):

$$P_{\text{BAD}} \propto \min(u_{01}, \log(\text{AraC})) + f_{\max} \quad (8)$$

when the $u_{01} + f_{max}$ is the highest output. Empirically, we observed that $f_{\max}$ can be estimated as:

$$f_{\max} = w_1 \cdot \log(\text{aTc}/\text{IDR}_4) + \text{const} \quad (9)$$

In Eq. 9, "Methods," the log-transformed perceptgene output is presented as a function of $\log(\text{IPTG}/\text{IDR}_3)$ (Supplementary Fig. 34a). In this analysis, we displayed the perceptgene output as a 2D plot, where the $x$-axis is IPTG concentration, and aTc is a constitutive parameter over IPTG values. Alternatively, $f_{\max}$ can be expressed as a function of IPTG if the preceptgene output is presented as a function of $\log(\text{aTc}/\text{IDR}_4)$. In our design (Fig. 1g), the input to $P_{\text{BAD}}$ activation function is a linear combination of weighted inputs (IPTG, aTc), with weights $n_3$ and $n_4$:

$$\log\left(\frac{\text{AraC}}{\text{AraC}_{\max}}\right) = n_3 \cdot \log(\text{IPTG}/\text{IDR}_3) + n_4 \cdot \log(\text{aTc}/\text{IDR}_4) \quad (10)$$

where $\text{AraC}_{\max}$ is the maximum protein level ("Methods," Perceptgene abstract model). Using the mathematical identity (Supplementary Notes, Smooth logical functions).

$$\min\{u_{01}, x + y\} = \min\{u_{01} - y, x\} + y \quad (11)$$

the empirical $f_{\max}$ observation from Eq. 9, "Methods," and defining $x = n_3 \cdot \log(\text{IPTG}/\text{IDR}_3)$ and $y = (n_4 - w_1) \cdot \log(\text{aTc}/\text{IDR}_4) + \log(\text{AraC}_{\max})$ for Eq. 11, "Methods," above, we obtain that the perceptgene computes:

$$\min\left\{u_{01} - \log(\text{AraC}_{\max}) - (n_4 - w_1) \cdot \log\left(\frac{\text{aTc}}{\text{IDR}_4}\right), n_3 \cdot \log\left(\frac{\text{IPTG}}{\text{IDR}_3}\right)\right\}$$
$$+ n_4 \cdot \log\left(\frac{\text{aTc}}{\text{IDR}_4}\right) + \text{const}_1 + \log(\text{AraC}_{\max}) - u_{01} \quad (12)$$

The term $\log(\text{AraC}_{\max}) - u_{01}$ is equal to $\log(B_2)$, where $B_2$ is the bias (Eq. 3.3.3, Supplementary Notes). The smooth minimum computation between $n_3 \cdot \log(\text{IPTG}/\text{IDR}_3)$ and $\log(B_2) - (n_4 - w_1) \cdot \log(\text{aTc}/\text{IDR}_4)$ can be extracted by subtracting the normalized $\log(GFP)$ output signal by $0.5 \cdot \log(\text{aTc}/\text{IDR}_4) - 0.4$ since $n_4 \approx 0.5$, and $\text{const}_1 + \log(B_2) = -0.4$.

Smooth maximum functions: When $P_{\text{BAD}}$ promoter is induced with high Arabinose levels, we can approximate its activity within the desired input dynamic range as a shifted and biased log-transformed positive rectifier activation function (Supplementary Fig. 32c):

$$P_{\text{BAD}} \propto \begin{cases} \text{constant} & \log(\text{AraC}) < u_{02} \\ \log(\text{AraC}) & \log(\text{AraC}) > u_{02} \end{cases} \quad (13)$$

Numerically, Eq. 13, "Methods," can also be approximated as (Supplementary Notes, Smooth logical functions):

$$P_{\text{BAD}} \propto \max(u_{02}, \log(\text{AraC})) + f_{\min} \quad (14)$$

with $u_{02} + f_{\min}$ signifying the lowest output. Empirically, we observed that $f_{\min}$ can be estimated as:

$$f_{\min} = w_2 \cdot \log\left(\frac{\text{AHL}}{\text{IDR}_5}\right) + \text{const}_2, \quad (15)$$

In our design (Fig. 2d), the input to $P_{\text{BAD}}$ activation function is a linear combination of the weighted input signals (AHL, aTc), with weights $n_5$ and $n_6$:

$$\log\left(\frac{\text{AraC}}{\text{AraC}_{\max}}\right) = n_5 \cdot \log(\text{AHL}/\text{IDR}_5) + n_6 \cdot \log(\text{aTc}/\text{IDR}_6) \quad (16)$$

("Methods," Perceptgene abstract model). Using the mathematical identity (Supplementary Notes, Smooth logical functions):

$$\max(u_{02}, x + y) = \max(u_{02} - y, x) + y \quad (17)$$

the empirical $f_{\max}$ observation from Eq. 15, "Methods," and defining $x = n_6 \cdot \log(\text{aTc}/\text{IDR}_6)$ and $y = (n_5 - w_2) \cdot \log(\text{AHL}/\text{IDR}_5) + \log(\text{AraC}_{\max})$, this yields a perceptgene that computes

$$\max\left(u_{02} - \log(\text{AraC}_{\max}) - (n_5 - w_2) \cdot \log\left(\frac{\text{AHL}}{\text{IDR}_5}\right), n_6 \cdot \log\left(\frac{\text{aTc}}{\text{IDR}_6}\right)\right)$$
$$+ n_5 \cdot \log\left(\frac{\text{AHL}}{\text{IDR}_5}\right) + \text{const}_2 + \log(\text{AraC}_{\max}) - u_{02} \quad (18)$$

The term $\log(\text{AraC}_{\max}) - u_{02}$ is equal to $\log(B_3)$, where $B_3$ is the bias (Eq. 3.8, Supplementary Notes). Hence, this perceptgene computes the maximum between two log-transformed analog numbers that are related to AHL and aTc, which is then offset by an analog value that is related to AHL only (Supplementary Fig. 36). The smooth maximum computation between $n_6 \cdot \log(\text{aTc}/\text{IDR}_6)$ and $\log(B_3) - (n_5 - w_2) \cdot \log(\text{AHL}/\text{IDR}_5)$ can be extracted by subtracting the normalized $\log(GFP)$ output signal by $0.46 \cdot \log(\text{AHL}/\text{IDR}_5) - 0.1$ since $n_5 \approx 0.46$ and $\text{const}_2 + \log(\text{AraC}_{\max}) - u_{02} = -0.1$.

## Strains, media, and chemicals

*Escherichia coli* 10-beta (#C3019H, New England Biolabs) was used for plasmid construction and all experiment assays. All liquid media used in the study was Luria-Bertani broth (LB, 10 g L$^{-1}$ tryptone, 5 g L$^{-1}$ yeast extract, and 10 g L$^{-1}$ sodium chloride) in liquid medium or agar supplemented with the appropriate antibiotics at the final concentrations of: Kanamycin (K1377, Sigma-Aldrich), 30 µg mL$^{-1}$; Chloramphenicol (C0378, Sigma-Aldrich), 34 µg mL$^{-1}$; Carbenicillin (10177012, Invitrogen), 50 µg mL$^{-1}$. The specifics of *E. coli* 10-beta include: araD139 D (araleu) 7697 fhuA lacX74 galK (W80 D (lacZ) M15) mcrA galU recA1 endA1 nupG rpsL (StrR) D (mrr-hsdRMS-mcrBC).

All chemicals used in the study are of the highest analytical grade. For preparation of stock inducers, powder of L-(+)-Arabinose (A3256, Sigma-Aldrich, 1.3 M), N-(3-Oxohexanoyl)-L-homoserine lactone (AHL, K3007, Sigma-Aldrich, 1 mM), Isopropyl-β-D-1-thiogalactopyranoside (IPTG, I6758, Sigma-Aldrich, 1 M) was dissolved in water. Stock solution of Anhydrotetracycline (aTc, 631310, Takara Bio, 20 µg mL$^{-1}$) was prepared using organic solvent such as ethanol.

## Plasmid construction and molecular cloning

All the plasmids in this work were constructed using basic molecular cloning techniques[79]. PCR was carried out with a Bio-Rad S1000™ Thermal Cycler. Oligonucleotides were synthesized by Integrated DNA Technologies (Coralville, IA). Restriction digestion enzymes were purchased from New England Biolabs (Beverly, MA) and Thermo Scientific FastDigest. Ligation were performed using T4 DNA Ligase (#M0202, New England Biolabs), and PCR was performed using Phusion High-Fidelity PCR kit (#E2621, New England Biolabs).

Synthetic DNA constructs were built using conventional subcloning using restriction digestion and ligation, Gibson Assembly and Site-directed mutagenesis and with the method chosen depending on their individual needs.

Parts are defined as promoters, RBSs, genes, terminators, origin of replication and antibiotic resistance. Manipulation of different parts of the same type was carried out using the same restriction sites; the origin of replication was cut with AvrII and SacI restriction enzymes, the gene was digested with Acc65I and BamHI restriction enzymes and the antibiotic resistance was cut with SacI and AatII/XhoI restriction enzymes.

To assemble multi parts we used the Gibson Assembly Master Mix (#E2611L, New England Biolabs) to join the DNA fragments (Ipswich, MA), following the manufacturer's instructions. The overlapping inserts were prepared by PCR amplifications using the Phusion High-Fidelity DNA Polymerase (#M0530L, New England Biolabs). For the purification of DNA, the Monarch Nucleic Acid Purification Kits were used (#T1030, New England Biolabs). Each assembly reaction contained approximately 250 ng of DNA fragments, followed by incubated at 50 °C for 60 min. For transformation, we used standard heat shock in *E. coli* 10-beta cells, followed by colony PCR screening on the next day. Selected colonies were grown overnight for miniprep (Qiagen, Hilden, Germany) and sent for standard sequencing (Macrogen Europe, The Netherlands) using appropriate primers. Mutations in the P$_{lux}$ promoter were performed using site-directed mutation (210518, Agilent QuickChange lightening), following the manufacturer's protocol. Random mutations were first performed in a simple circuit, P$_{lux}$-GFP-P$_{lacO}$-luxR. Transfer functions of mutated colonies were characterized. The mutations with desired characteristics were then selected for sequencing and integrated with the other parts in this work. All synthetic parts used in this work are listed in Supplementary Notes, List of biological parts used in this study. Supplementary Notes, Plasmid Maps, provides details regarding plasmid maps. Table in Supplementary Notes, List of strains used in this study, provides the list of strains used in this study. The references in Supplementary Notes, Supplementary References, provide details regarding the origin of the plasmids.

## Circuit characterization

Overnight cultures of *E. coli* strains were grown from frozen glycerol stocks at 37 °C, in a Shel Labs SSI5 shaking incubator at 250 r.p.m., in 5 ml of LB with appropriate antibiotics: Carbenicillin (50 µg mL$^{-1}$), Kanamycin (30 µg mL$^{-1}$), Chloramphenicol (34 µg mL$^{-1}$). The inducers used were Arabinose, IPTG, aTc and AHL (Sigma Aldrich). Overnight cultures were diluted 1:100 into 5 mL fresh LB with appropriate antibiotics and were incubated at 37 °C, 250 r.p.m. for 30 min. Cultures (200 µL) were then moved into 96-well plates, combined with inducers, and incubated for 4 h and 20 min in a microplate shaker (37 °C, 500 r.p.m.) until they reached an OD$_{600nm}$ ~ 0.4–0.6. Then, the fluorescence and scattering of bacterial cultures were analyzed through flow cytometry analyzer.

The flow cytometry analyzer voltages were adjusted using CyExpert 2.2 software so that the maximum and minimum expression levels could be measured with the same voltage settings. Thus, consistent voltages were used across each entire experiment. The same voltages were used for subsequent repetitions of the same experiment. GFP was excited with a 488 nm laser, and mCherry was excited with a 561 nm laser.

In all experiments, 10,000 events have been obtained and the fluorescence and forward and side scattering were taken using the CytExpert 2.2 software (Supplementary Fig. 256). The geometric median of the gated fluorescence distributions was calculated using MATLAB. Fluorescence values were based on geometric medians of each population from three experiments and is reported here as the fluorescence value of a sample in arbitrary units (arb. units). The flow cytometry data for one representative experiment is provided in Supplementary Notes, FACS Data, (Supplementary Fig. 256).

## Reporting summary

Further information on research design is available in the Nature Research Reporting Summary linked to this article.

## Data availability

The flow cytometry experiments generated during the current study are available on GitHub with https://doi.org/10.5281/zenodo.7041620. Source data are provided with this paper. Zenodo ULR: https://zenodo.org/record/7040614#.YxmnkXbP2Uk. Source data are provided with this paper.

## Code availability

The algorithm codes that are used in this study are available on GitHub with https://doi.org/10.5281/zenodo.7041620. Zenodo ULR: https://zenodo.org/record/7040614#.YxmnkXbP2Uk.

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

## Acknowledgements
The authors would like to thank Dr. Ximing Li, Roee Samul, and Raghed Abu Sinni from the Technion- Israel Institute of Technology. This work was partially supported by the Neubauer Family Foundation, Israel Science Foundation (ISF) [1558/17], European Union's Horizon 2020 Research and Innovation Programme FET-Open NEU-Chip under grant agreement No. 964877, and the Technion's Lorry I. Lokey interdisciplinary Center for Life Sciences and Engineering. Support for R.W. was provided by the Defense Advanced Research Projects Agency under contract no. W911NF-21-2-0109.

## Author contributions
L.R., L.D., R.W., M.H., and R.D. designed the study. L.R. and M.H. performed experiments and collected data. L.D. and R.D. invented the neuromorphic circuit motifs. L.R. and R.D. associated models and simulations. R.D. and R.W. wrote the manuscript. All authors analyzed the data, discussed results. and wrote the manuscript.

## Competing interests
The authors declare no competing interests.
