## [Peer Review File · Nature Communications]

Reviewers' Comments:

Reviewer #1:

Remarks to the Author:

I recommend publication of the manuscript in Nature Communications as it is highly novel, technically rigorous, and of broad and timely interest for the general readership.

For years, the focus in synthetic biology has been to implement digital-like computing, which is robust but requires a prohibitively increasing number of elements to compute meaningful functions, and adds a major perturbation to the host cell.

Daniel and his lab now change this paradigm toward neural-network computation that can be digital-like as well as continuous analog, with a minimal number of components and using simple design principles. The manuscript is a milestone in biological computing, synthesizing the fact that biochemical interactions operate in the domain of the logarithm of concentrations, with neuromorphic computing. Cleverly, the authors do so by a simple concept of implementing a mathematical perceptron using regulatory auto-negative and auto-positive feedback loops to create their perceptgene. This enables the design of a plethora of genetically encoded computational schemes not possible till now, including multilayer perceptgene networks and backpropagation algorithm for three input majority function, and data converters based on neural network principles.

The manuscript is solid and rigorous, with ample data, theoretical derivations and simulations to support their claims.

Reviewer #2:

Remarks to the Author:

The manuscript describes the dynamics of rather simple genetic networks in analogue terms. The idea is interesting, as analogue information-processing seems closer to molecular signalling than digital-like computing. This is worth investigating further. However, I find it extremely difficult to read. Overcomplicated explanations and convoluted figures. The take-home message is not clear. Is it about information flows? About circuit topologies? Are authors suggesting new mathematical interpretations? Are they describing novel circuits? The fact that the paper lacks of structure (no Intro, Methods, Results, etc) or even a References section can surely lead to confusions. If authors are providing a 300 pages long SI, why convoluting the main text so much?

Here a few comments that could help improving the text, in my opinion.

- Please identify the story—the message.
- Introduce neuromorphic computing.
- The majority gate is simply not working well at low arabinose concentrations. Please clarify.
- Figures need more detail, while removing superfluous information. For instance, in Fig1B IPTG and aTc should also inhibit the other two interactions. In Fig2A lines don't match. In Fig2F the equations in the axis are not helping get the idea. Fig3G is difficult to read—is the formulation needed? This is not even explained in the main text...
- I do not understand why using the perceptron as a metaphor. The perceptron is an algorithm that learns. It can be trained. The suggested "perceptgene" does not learn, nor can be trained. This is a major point, in my opinion.
- I found the negative rectifier bit particularly difficult to read. Perhaps authors could move some of the more technical aspects to SI or a Methods section and explain clearly what they mean.
- I fail to see how the "perceptgene" classifies anything.
- AND gate designs are not that different from the "perceptgene" (if at all). Could authors please explain why this network is so different?
- Sentences like "the power-law functions compute the weighted inputs" are, to me, confusing. Authors are talking about a positive regulation.
- P7L16 says "perform different computational tasks by modifying various parameters". What are those parameters? What tasks?

- P16L6 "the best performance was observed for a lower level of arabinose". According to the picture, I do not agree. Please clarify.
- How is that Fig4H is a switch?
- In P13 authors talk about rows in Table 1. This is so confusing. I do not understand the table.

Reviewer #3:

Remarks to the Author:

The manuscript by Rizik et al. describes a framework for engineering genetic circuits that implement neural-like network computations in *E. coli*. The circuits are based on the classic ligand-inducible transcription factors LacI, TetR, AraC, LuxR. The authors integrate these transcription factors into small feedback-containing circuits (perceptgenes) that perform logarithmic computations - similar to previous work on analog gene circuit computation. The critical parameters of weight and bias are manipulated via an assortment of ligand concentration, transcription factor, and promoter modifications. Multiple perceptgenes are inter-connected via additional regulated promoters to implement genetic majority computation and analog-to-digital conversion (both systems convert the presence of ligands to gene expression output patterns at steady state). The experimental work is guided and interpreted by theory.

The implementation of neural network computations in gene regulatory networks is a fascinating result. The major limitations of the manuscript are that the authors rely upon idiosyncratic observations about the performance of their gene regulatory parts to implement critical processes such as the tuning of weights and biases that dictate the function of the networks, and the general design principles by which one would vary these variables in genetic circuits is not clear. Thus, most of the future advancements on this work will likely require researchers 'starting from scratch' with other classes of parts with more programmable properties and re-conceiving of approaches to tune weights, biases, etc.

I also have a number of other concerns. Specific comments below:

Major comments

1. A major limitation of the paper is that the weights of inputs cannot be systematically engineered. Rather, the authors usually empirically determine the weights of commonly used gene regulatory parts, or small libraries or random mutants derived from those parts, and then select parts with appropriate weights to implement specific neural computations. Furthermore, inducer (arabinose) concentration is used to alter the weight of AraC. This appears to work due to the biochemical details of the AraC/P_{bad} system and is not likely to generalize to other inducible promoters. Finally, arabinose also alters the bias of AraC, which confounds the experiments. Overall, these limitations appear to significantly constrain the breadth of neural network operations that can be implemented using the approach in the paper. The author should add a discussion of these limitations of their approach.
2. The fluorescent protein reporter data are reported in a highly inconsistent manner throughout the paper. For example, the data demonstrating the perceptgene power law and multiplication sub-circuit function is plotted in normalized units in Figures 1C and D, but in non-normalized units in Figures S9.1 and S9.2. The lack of standardization of units makes it difficult to compare the performance of different systems or the same system in different experiments or genetic contexts.
3. The authors claim that the P_{bad} promoter exhibits a log-transformed sigmoidal response (page 9, paragraph 2 and Fig. S2.9). Based on the description and model of perceptgene function that the authors provide elsewhere in the manuscript, one would assume that the relevant response of P_{bad} is its response to the concentration of AraC. However, the experiment performed in Fig. S2.9 does not provide the response of P_{bad} to AraC concentrations. In particular, the authors use AHL to induce expression of a degradation tagged version of AraC from a positive autoregulatory promoter (Fig. S2.9a). The relationship between AHL and AraC levels is not measured, and it should not be assumed that this relationship is linear, especially due to the positive autoregulation and degradation tag in this characterization circuit. On a related note, how do the authors arrive at log(AraC) values that they use to show negative rectifier function in Fig. S3.1? It does not appear

they have the necessary data to conclude negative rectifier function.

4. The origin of the perceptgene model (Supplementary Information, Section 1 - Box 1) is not described. Thus, it is not clear how this model facilitates the design or understanding of the performance of a perceptgene. For example, why is perceptgene bias determined by "translation and transcription rates, mRNA/protein half-lives, rate of cell growth, binding affinities in protein-protein or protein-DNA reactions". How do such a large number of disparate process combine to result in a bias? How would one design or modify a perceptgene to modify the bias given this complex entanglement of processes?

5. The description of the perceptgene model is poor, making it difficult to follow. For example, in describing equation (I) the authors state "Where X_i is the input concentration, K_{mi} is dissociation constant, or input dynamic range or normalization". Is K_{mi} a dissociation constant? If so, of what? Or is it an input dynamic range? Or is it a normalization term?

6. The scales and units differ between the experimental and simulated sub-circuit characterization data in Figures 2B and C. This makes it difficult to compare experimental and simulation results. The authors should account for these discrepancies.

7. The analog-to-digital converter data (Figure 4) is a strength of the paper. I believe that this result shows the implications of the framework in a particularly compelling way.

8. On page 21, the authors seem to imply that gene regulatory parts with steep transfer functions (high Hill coefficients) would be beneficial to their framework. However, if or why this is the case is not made clear. The authors should clarify this section of the discussion and enumerate the benefits of such digital-like parts if they would indeed be beneficial.

9. The authors state: "We anticipate that the new framework described here constitutes a first step towards implementing supervised machine learning optimization algorithms in individual living cells". This is certainly an intriguing claim. However, the implications of achieving supervised learning in cells, and the technical limitations that must be overcome are not described. This information should be added, or the statement should be removed.

10. Why is mCherry used as a reporter gene in Figure 2A-C, when GFP is used throughout the rest of the paper? This peculiar choice compounds with the sporadic use of normalization to make it difficult to interpret the performance of circuits throughout the paper.

Minor comments:

1. The schematics depicting the gene regulatory network designs are non-standardized and difficult to interpret. For example, in Figure 1, promoters are represented as rightward facing block arrows that occupy the entire height of the depicted DNA. This representation is commonly used to depict open reading frames. Open reading frames are depicted as the absence of any glyph on the DNA, which is also unusual. Additionally, rather than depicting a specific origin of replication or a category of origins or replication using a glyph, the authors write "LCP", "MCP", or "HCP" for low-medium- and high-copy plasmid in text next to the DNA. Furthermore, the style of the depictions changes within the document (e.g. Fig. S.2.2). At a minimum, the schematics should be standardized. I recommend the authors follow current SBOL visual standards.

2. The words "LacI binding site" should be replaced with the sequence of the LacI O1 binding site in Fig. S2.5. The label "O1" beneath the LacI binding site region is sufficient to indicate that the sequence is a LacI operator.

3. What is the relevance of the fact that the perceptgene responses are stable between 4.5 and 10 hours? Do they fail after 10 hours? In any case, this number is likely to be limited by batch culture growth conditions and adding the inducers at only one time point.

Synthetic neuromorphic computing in living cells

Point-by-point response to the reviewers' comments

July 9, 2022

We thank all reviewers for the comments. We have devoted considerable efforts to carrying out new experiments. We have included a new figure (Fig. 5), added a new analysis, modified the narrative, and expanded the discussion in many places (**changes are marked in blue**). Our Supplementary Information has also been expanded to provide additional information regarding our system. *Reviewer comments are italicized. Our answers are in bold.*

Reviewer #1 (Remarks to the Author):

I recommend publication of the manuscript in Nature Communications as it is highly novel, technically rigorous, and of broad and timely interest for the general readership. For years, the focus in synthetic biology has been to implement digital-like computing, which is robust but requires a prohibitively increasing number of elements to compute meaningful functions, and adds a major perturbation to the host cell.

Daniel and his lab now change this paradigm toward neural-network computation that can be digital-like as well as continuous analog, with a minimal number of components and using simple design principles. The manuscript is a milestone in biological computing, synthesizing the fact that biochemical interactions operate in the domain of the logarithm of concentrations, with neuromorphic computing. Cleverly, the authors do so by a simple concept of implementing a mathematical perceptron using regulatory auto-negative and auto-positive feedback loops to create their perceptgene. This enables the design of a plethora of genetically encoded computational schemes not possible till now, including multilayer perceptgene networks and backpropagation algorithm for three input majority function, and data converters based on neural network principles.

The manuscript is solid and rigorous, with ample data, theoretical derivations and simulations to support their claims.

We are grateful for the kind words and support for this work.

Reviewer #2 (Remarks to the Author):

1. The manuscript describes the dynamics of rather simple genetic networks in analogue terms. The idea is interesting, as analogue information-processing seems closer to molecular signalling than digital-like computing. This is worth investigating further. However, I find it extremely difficult to read. Overcomplicated explanations and convoluted figures. The take-home message is not clear. Is it about information flows? About circuit topologies? Are authors suggesting new mathematical interpretations? Are they describing novel circuits?

We thank the reviewer for the valuable feedback, which helped us further clarify and emphasize the contributions of our work. We provided comprehensive responses to the points raised by the reviewer as below, and we believe that the take-home message is more apparent in the revised manuscript.

The manuscript presents a new computational framework inspired by neuronal networks to design gene circuits. The proposed framework contains information flows and a new mathematical representation to encode biological information. Based on the proposed neuro-inspired computing framework, our study suggests new network topologies such as the one that computes the three-input majority (Fig. 3), and new circuits such as the 2-bit analog-to-digital converters and ternary switch (Fig. 4). Such topologies and circuits have not been demonstrated in synthetic biology.

Our selection of circuits to design and build was based on the following:

- 1. To prove that neuromorphic computing principles and algorithms (such as gradient descent and backpropagation, Fig. 3), can be achieved in single living cells by transforming concepts from neural networks to genetic regulatory networks.**
- 2. To construct synthetic gene circuits that perform complex computation with minimal requirements in terms of computational devices and host cell resources (Fig. 4).**
- 3. To efficiently fine-tune and repurpose gene circuit functions (Fig. 4 and 5).**

Three of the circuits we decided to build, min/max/avg, are considered to be fundamental building blocks for neuromorphic computing (Figs. 1 and 2). The other three, majority / 2-bit ADC / ternary switch, demonstrate multi-layered neuronal circuits (Figs. 3 and 4). In the revised manuscript, we provide a new gene circuit where its computation can be programmed and modified from OR to AND (Fig. 5) by modulating the perceptgene weight using small molecules.

In terms of appreciating the min/max/avg functions, it is important to recognize that to the best of our knowledge, only “hard” (i.e. discrete) minimum and maximum functions have been demonstrated in synthetic biology. Hard functions are ones that operate using binary AND/OR logic, where each bit has two logic states. In terms of these hard functions, AND implements binary min,

while OR implements binary max. The average function cannot be implemented with an individual single-bit binary logic gate, but would rather require digitization of input and output signals and very complex multi-device logic. By contrast, soft functions operate in the analog domain. Our single perceptgenes implement single-device analog min/max/avg computations whereby the single devices transform analog input signals into output values that remain in the analog domain. These operations have not been demonstrated in synthetic biology!

Our multi-layer functions also represent significant progress over existing efforts in synthetic biology. First, our majority function (Fig. 3) demonstrates neuromorphic modularity because the three-input majority function is built from two perceptron layers. Second, our three-input majority function uses fewer synthetic parts than previous digital designs (Nielsen *et al.*, 2016). Our three-bit majority function comprises 15 biological parts (i.e., promoters and genes) in comparison to 22 parts. Third, with this neuromorphic architecture, we minimized error by modulating the weights of $P_{BAD}/AraC$ and $P_{lux}/LuxR$ in a manner similar to the backpropagation algorithm. This optimization approach could not have been performed for the existing digital circuit design.

Our other two multi-layer neuromorphic circuits also provide innovation beyond existing approaches. To the best of our knowledge, we are the first to demonstrate a 2-bit analog-to-digital converter (ADC). In general, analog-to-digital converters take a graded signal as input, partition the analog input into several consecutive ranges that cover the entire input range and assign a digital value to these ranges sequentially. Representing this digital value requires multiple bits if more than two regions are specified. A 1-bit ADC partitions the input range into two, and the output is then a single bit with a value of either 0 or 1. A 2-bit ADC partitions the input range into four, with an output that requires two bits to represent each of the four consecutive ranges, namely 00, 01, 10, and 11.

The implementation of our third multi-layered neuromorphic circuit, the ternary switch, demonstrated the ease of converting one computing task to another function (Fig. 4H). We started with the 2-bit ADC circuit and increased activation of the LSB perceptgene using higher Arabinose, which corresponds to increasing the MSB input weight into the LSB computation. Such ability to change neuronal network parameters and achieve new functions is an important component for ultimately implementing learning algorithms using gene circuits. To the best of our knowledge, we are the first to implement a ternary switch. In terms of computational flexibility, we also provided a new experiment demonstrating perceptgene circuits' ability to repurpose their logic computational functions by changing weights and biases. We showed that by increasing mainly the weights of a

perceptgene from low to high, its computation can be modified from an AND-like manner to an OR-like manner (Fig. 5).

2. The fact that the paper lacks of structure (no Intro, Methods, Results, etc) or even a References section can surely lead to confusions.

We thank the reviewer for the suggestion. The revised manuscript includes the following sections Introduction, Results, Discussion, Methods and References.

3. If authors are providing a 300 pages long SI, why convoluting the main text so much?

We thank the reviewer for the suggestion. In the revised manuscript, we moved the mathematical descriptions to Methods section.

Here a few comments that could help improving the text, in my opinion.

4. Please identify the story—the message.

Nonlinear computation is prevalent in biological systems across scales such as neuron activity, cell differentiation, and the vertebrate adaptive immune system; where the invasion of pathogens triggers a series of actions from multiple cell types to protect the organisms. Hence, platforms that use nonlinear functions are more suitable for engineering events and behaviors mimicking natural biological systems that involve decision-making, in a manner that is difficult to achieve with digital or analog platforms alone. Exploiting design based on nonlinear computation will enable more complex and more robust applications in synthetic biology. However, implementing design principles of nonlinear computations in physical and biological systems is challenging. One promising approach is neuromorphic engineering that uses abstract models of neural networks to design systems is one form of nonlinear computation. Recently, it has been shown that neuromorphic principles can be applied to engineer electronic circuits that can solve sophisticated tasks by combining analog signals with nonlinear activation functions. The unique properties of neuromorphic circuits and their use in artificial intelligence and machine learning are revolutionizing how we approach many complex computational problems. We anticipate that a similar revolution will transform how we design biological circuits. With neuromorphic computing, we can, in principle, optimize, *in situ*, the design parameters of biological circuits to desired functional criteria using the powerful backpropagation learning algorithms of machine learning in living cells.

To efficiently implement neuromorphic computing in a single cell, we created the perceptgene model that comprises weights, bias, and activation functions. We first mapped these parameters to biological factors (Methods, BOX I), and then built a programmable platform that allows us to build perceptgene networks with different behaviors. In general, weights and biases in our neuromorphic circuits are determined by several factors that include, but are not limited to:

- (1) Hill coefficients of small-molecule inducers serving as perceptgene inputs (Fig. 1 and 2),
- (2) number and sequence of transcription factor binding sites (Fig. 1 and 3),
- (3) circuit topology such as the regulation of negative feedback (Fig. 5G, fig. S2.8.1, fig. S8.3),
- (4) circuit topology such as the regulation of incoherent feedforward strength (fig. S8.5),
- (5) transcription factors that competitively inhibit expression via steric hindrance (fig. S8.6),
- (6) transcription factor sequestration via protein-protein interactions (Fig. 5, fig. S8.7),
- (7) operator sequence that controls binding affinity of transcription factor (fig. S8.8 and S8.10),
- (8) protein structure (*e.g.*, dimerization and cooperativity),

Our abstract begins by pointing out similarities between neural and biological regulatory networks in cells. It then discusses how a simplified network model using interconnected perceptrons can be used to capture the information processing and computational capabilities of neural networks and how this model inspired a revolution in electronics and software algorithms. We then suggest that artificial neuronal networks are particularly suitable for overcoming several important challenges in designing synthetic gene circuits. Of note, implementing artificial neural networks using gene regulation required an essential transformation: the creation of perceptron-like devices (perceptgenes) that operate in the logarithmic domain.

To clarify the points raised above, we highlight the role of nonlinear computation of biological systems in the introduction section (page 3, line 21, and page 4, line 1).

Furthermore, we moved BOX I to the Methods section and added additional content describing the required steps to develop the perceptgene abstract model. We started with the formalization of a hybrid promoter having multiple inputs, and then we defined an ordinary differential equation for gene expression. In the end, we emphasize the role of translation/transcription rates, mRNA and protein half-lives, and cell growth rate.

Finally, we added a sentence in beginning of each section to highlight the main message

5. Introduce neuromorphic computing.

Neuromorphic computing is a terminology that originated from electrical engineering, with the goal to partially replicate nervous system-like functions, including perception, motor control, and cognition, using silicon-based circuits and following algorithms developed for these systems. In our context, synthetic neuromorphic computing in living cells can be considered a way to engineer biological systems using concepts from neuronal networks.

In this study, we developed a new abstract model and biological design, “perceptgene,” that allows us to efficiently implement neuromorphic computing circuits in living cells. The perceptgene and the

perceptron, the basic computing unit in artificial neural networks, have similar structures (weights, bias, and activation functions). The only difference between the both models is that the operating domain of the perceptron is linear, and the perceptgene is logarithmic. Neuromorphic engineering in living cells has the potential for

1. Studying the architectures of gene circuits and networks in similar to how neuromorphic engineering impacts neuronal circuits.
2. Implementing sophisticated computing functions that are hard to achieve with the digital framework, i.e., requires a large number of synthetic parts. Examples are min/max/avg functions that cannot be implemented with an individual single-bit binary logic gate by contrast to our single perceptgenes in Figures 1 and 2.

In addition, we showed that our three-bit majority function comprises 15 biological parts (i.e., promoters and genes) in comparison to 22 parts. Our other two multi-layer neuromorphic circuits also provide innovation beyond existing approaches. To the best of our knowledge, we are the first to demonstrate a 2-bit analog-to-digital converter (ADC). In general, analog-to-digital converters take a graded signal as an input, partition the analog input into several consecutive ranges that cover the entire input range, and assign a digital value to these ranges in a sequential manner. Representing this digital value requires multiple bits if more than two regions are specified. A 1-bit ADC partitions the input range into two, and the output is then a single bit with a value of either 0 or 1. A 2-bit ADC partitions the input range into four, with an output that requires two bits to represent each of the four consecutive ranges, namely 00, 01, 10, and 11. Two recent synthetic biology publications have discussed the notion of analog-to-digital converters. In one publication (Muller *et al.* 2017), 1-bit analog-to-digital conversion was used to quantize extracellular inputs (including dihydrojasnone and eugenol) each into single bit values, and then these were combined into several 2-input logic functions (AND, OR, NOR) still operating with single bit output. The information flow between the analog parts and logic functions is carried by cell-to-cell signaling molecules, by contrast, our computations are performed in single cells. In another publication (Rubens *et al.* 2016), a single analog input (H_2O_2) was partitioned into three consecutive ranges, and three separate 1-bit outputs (GFP, RFP, and BFP) were used to indicate which of three ranges was detected. Hence, for a given analog input value, one of these digital outputs is high while the other two are low. In conventional ADC circuit design, these three 1-bit outputs are then combined via a second stage digital logic circuit (comprising three 2-input logic gates: one XOR and two AND gates) to create a 2-bit digital representation of the analog input signal. Hence, this work represents only the first stage of a 2-bit ADC, but not the (very important) second stage. In

terms of biological circuit elements, they used seven transcription units. We estimate that it would require 6-8 additional transcription units to implement their second stage of the 2-bit ADC, which would require a total of 15 transcription units if it was built. In comparison, ours is a fully functional 2-bit ADC implemented using only five transcription units. Besides minimizing the size of the circuits, our perceptgene networks also operate with low expression levels, mainly in order to maintain low bias levels. In contrast, digital systems often attempt to operate with significant noise margins, and hence high expression levels for ON values. This latter point is further elaborated on in the main narrative.

3. **Implementing biological intelligence.** Recently the synthetic biology community has been increasingly interested in creating “smart cells” for a variety of applications, as was articulated in “Computer logic meets cell biology: how cell science is getting an upgrade” (Savage *et al.*, Nature 2018). The article claims that new approaches that combine intelligence and the regulation of biological systems are essential for achieving new objectives. The term intelligence has different levels, starting from optimization, reconfigurable, supervised learning, trainable, and ending with unsupervised learning or self-adaption. One possible way to build intelligent systems is by implementing perceptron models. In this study, we showed that three input majority neuromorphic circuit (Fig. 3) is reconfigurable and trainable via learning algorithms that optimize desired behavior efficiently (e.g., reduce error). With this neuromorphic architecture, we minimized error by modulating the weights of P_{BAD}/AraC and AHL/LuxR in a manner similar to the backpropagation algorithm. This optimization approach could not have been performed for the existing digital circuit design. The implementation of our third multi-layered neuromorphic circuit, the ternary switch (Fig. 4), demonstrated the ease of converting one neuromorphic computing to another. Specifically, we started with the 2-bit ADC circuit and increased activation of the LSB perceptgene using higher Arabinose concentration, corresponding to increasing the MSB input weight into the LSB computation. The ability to change neuronal network parameters and achieve new functions is essential for ultimately implementing learning algorithms using gene circuits. To the best of our knowledge, we are the first to implement a ternary switch. To highlight this point, we provided a new experiment (Fig. 5) demonstrating how we can program the logic computation of a two-input perceptgene circuit from AND to OR logic gates by increasing the perceptgene internal weight using an external AHL induce. In terms of more complex computation, we also provided a new theoretical analysis of a 2-bit Full Adder implementation comparing the neuromorphic versus digital approaches (Main text, Page 22, lines 7 to 9, Supplementary Information, Section 8).

Finally, we also showed that logarithmic algorithms could outperform linear algorithms when implemented in gene circuits. Computation in the logarithmic domain allowed us to implement neuromorphic circuits by readily achieving desired weight and bias behaviors. The logarithmic domain is also more appropriate for attenuating the effects of typical fluctuations in protein expression levels, and hence provides a more resistant platform for neuromorphic computing in a gene regulation context. The essence here is the reliance on fold-change regulation, as opposed to absolute-change regulation, with the former being more appropriate for genetic circuits (as previously articulated in the community, e.g. by Uri Alon). In comparison, a linear genetic implementation would have been based on less reliable parts, and hence necessitated a more complex design to achieve the same performance. This point is discussed in details in the main narrative (Discussion, Page 19, lines 20 to 23, and page 20, lines 1 to 14) and a detailed technical discussion of these properties in the revised supplementary information (Supplementary Information, Section 11). To clarify the points raised above, we define neuromorphic computing in the introduction section (page 4, lines 20 to 24, page 5: lines 1 to 7).

6. The majority gate is simply not working well at low arabinose concentrations. Please clarify

We focus on building perceptgene with soft classification using a sigmoid function, rectifiers function, and linear function rather than hard classification using a step function. To observe a majority function with a hard classification response when the output has only two-discrete binary states, we can connect the final layer of the proposed majority function with a synthetic part that has a sharp response (steep input-output transfer function). For example, we can replace the GFP of the circuit in Fig. 3B with recombinase protein [P Siuti *et al.* 2013] and may observe a hard majority function. In the revised manuscript, we used the “soft” majority function to highlight that our majority is smooth. We expanded the discussion section to include this point (page 21, lines 20-22). A further description for implementing a hard majority using recombinase is provided in the supplementary information section 5, Fig. S5.10.

7. Figures need more detail, while removing superfluous information. For instance, in Fig1B IPTG and aTc should also inhibit the other two interactions. In Fig2A lines don't match. In Fig2F the equations in the axis are not helping get the idea. Fig3G is difficult to read—is the formulation needed? This is not even explained in the main text...

We thank the reviewer for pointing this out:

- **Inducers that inhibit other interactions were added in all relevant figures.**
- **We rebuild Fig. 2A**

- We plotted Fig. 2F in the form of three-dimensional space to clarify the computation process. In each figure, we added plots that theoretically compute the maximum/minimum/average operations between the analog numbers of the x-axis and y-axis.

8. I do not understand why using the perceptron as a metaphor. The perceptron is an algorithm that learns. It can be trained. The suggested “perceptgene” does not learn, nor can be trained. This is a major point, in my opinion.

We thank the reviewer for pointing this out. Nowadays, the perceptron frequently is used for implementing systems that support machine learning; however, in the early days of ANNs, several studies showed that perceptron could also be used for signal processing and computation, combining analog signals with decision-making capabilities (Neural networks: a systematic introduction / Raúl Rojas; foreword by Jerome Feldman. - Berlin; New York: Springer-Verlag, c1996, chapters 2 and 13). For example, analyses of feedforward ANNs have suggested using the perceptron for designing logic functions. The recurrent ANNs developed by Hopfield use the perceptron to build data converters and solve traveling salesman optimization by minimizing the network's energy function (Hopfield et al. Biological cybernetics 1985). Our theoretical analysis in Supplementary sections 5 and 7 demonstrated that perceptgene networks can be applied to implement three-input majority and data converters. In both circuits, we solved linear equations to determine weights and biases for optimizing network behavior. In addition, in our initial submission, we developed gradient descent and backpropagation algorithms suitable for perceptgene networks and gene circuits. We used these algorithms (Fig. 3G) to optimize the majority performance (Fig. 3H) by modulating $P_{BAD}/AraC$ using different Arabinose concentrations and $P_{lux}/LuxR$ by running random mutations on the LuxR operator. In the revised manuscript, we provided a new gene circuit that its computation can be modified from OR to AND (Fig. 5) by modulating the weights of $P_{exs}/ExsA$ via transcription factor sequestration using AHL molecules. We also included several biological factors and mechanisms that can be used to modulate weights and biases of the perceptgene in gene circuits (Supplementary Information, Section 8) that allow us to perform training conceptually.

9. I found the negative rectifier bit particularly difficult to read. Perhaps authors could move some of the more technical aspects to SI or a Methods section and explain clearly what they mean.

We appreciate the reviewer for the suggestion. In the revised manuscript, we moved the mathematical equations of rectifiers to the methods section (Methods, BOX II). We also simplified our narrative to clarify the smooth computation.

10. I fail to see how the “perceptgene” classifies anything.

The perceptron and perceptgene models can perform hard classification with a response of two-discrete binary levels (low/high) or soft classification for sub-groups. Hard classification requires an activation function with a step response, while soft classification uses activation functions with nonlinear (smooth) responses. While hard classification is useful for digital computation, soft classification is useful for analog and neural-like computation executing sophisticated functions. This study focuses on building perceptgene with soft classification using nonlinear activation functions such as sigmoid and rectifiers. In the revised manuscript, we illustrate Fig. 1 and 2 in the representation of three-dimension space to demonstrate the non-linearity of these circuits. We also evaluated the non-linearity degree for the perceptgene circuits by computing the

$$\text{NonLinearity} = \frac{\text{Hill-coefficient}}{\log(\text{Maximum Fold Change})/\log(\text{Input Range})}$$

The denominator $\log(\text{Maximum Fold Change})/\log(\text{Input Range})$ is equal to the slope of a linear line that spans between the (Y_L, β) and (Y_H, I) points at the log scale, where the term $\log(Y_H/Y_L)$ equals the input dynamic range. Therefore, the NonLinearity degree is computed as the ratio between two slopes – the slope of an input-output transfer function and the slope of a linear line with the same input dynamic range and maximum fold change of the target function (Supplementary Information, Section 1.6). For a linear line, the Hill – coefficient and $\log(\text{Maximum Fold Change})/\log(\text{Input Range})$ equal, and therefore, the NonLinearity=1. For step function, the Hill – coefficient is much larger than $\log(\text{Maximum Fold Change})/\log(\text{Input Range})$ and therefore, the NonLinearity \gg 1. The theoretical analysis of NonLinearity is provided to the Supplementary information section 1.6.

Our experimental results showed that the Nonlinearity degree increased by more than three folds when a sigmoid activation function was added (comparing Fig. 1B versus Fig. 1G) and increased by two folds when a positive rectifier activation function was added (comparing Fig. 2A versus Fig. 2D). The Nonlinearity for perceptgene with a linear activation function (Fig. 2H) equals one. These results are summarized in Table 1, added to the revised manuscript.

Furthermore, in computation and signal processing, there are several advantages to using soft classification rather than hard classification. For example, our theoretical analysis showed that the three-input majority function can be implemented with a single layer of a step activation function (e.g., the three input weights are equal to one, and the threshold equals 1.5). However, biologically implementing an activation function (e.g., a promoter with three different binding sites) with three inputs and obtaining the desired parameters (weights and bias) is often challenging. Alternatively,

we initially thought to implement a three-input majority function by cascading two perceptrons, each having two inputs. From our analysis of such a network, only perceptrons with a sigmoid activation function can satisfy the mathematical constraints on the design parameters to implement a three-input majority function. Therefore, the only feasible way to build a genetic circuit in E.coli that computes the majority of three inputs is by using two layers of a two-input sigmoid activation function, as was implemented in this study (Fig. 3). We added this discussion and the theoretical analysis to the Supplementary Information, Section 5, Table S5.3, and Table S5.4.

Importantly, the 2-bit analog-to-digital converter in Fig. 4 and the 2-input logic gates in Fig. 5 demonstrated perceptgene networks with hard classification.

11. AND gate designs are not that different from the “perceptgene” (if at all). Could authors please explain why this network is so different?

The AND logic gate is a specific case of perceptgene design. We can implement an AND gate using a perceptgene when the inputs weights or the bias values are low. By increasing the input weights from low to high, the perceptgene computation can be modified from an AND to an OR logic gate (Fig. 5). Such a feature can not be achieved using a digital design. This new experiment is provided in the discussion section of the revised manuscript.

In the new experiment, we induced the neuromorphic genetic circuit by a small molecule that controls transcription factor sequestration via protein-protein interactions (Fig. 5A). Specifically, we selected ExsD to shunt ExsA from binding P_{exs} promoter, which activates the GFP expression. In our design, we controlled ExsD by AHL: when AHL is low, the internal weights are high, and when AHL is high, the internal weights are low (Fig. 5C). We built a two-input perceptgene circuit using the combinatorial promoter ($P_{\text{lacO1/tetO}}$) and auto-negative feedback loops encoding LacI and TetR as shown in Fig. 1B. In the new design (Fig. 5D), we connected the power-law and multiplication function to ExsA/ P_{exs} activation function by encoding ExsA downstream of $P_{\text{lacO/tetO}}$. To control the internal weight, we induced ExsD by AHL using a weak P_{lux} mutant that broadens the AHL dynamic range. We induced the gene circuit with (1) AHL=0, which led to a high internal weight observing GFP signal with an OR-logic gate manner (Fig. 5E), (2) AHL=0.34 μ , which led to a low internal weight observing GFP signal with an AND-logic gate manner (Fig. 5F). To improve the accuracy of the OR circuit and get distinct low and high outputs (Fig. 5G), we increased the input weights by enhancing the strength of the auto-negative feedback loops. In the modified design, we constructed the auto-negative feedback loops of P_{lacO1} and P_{tetO} in medium plasmid copy numbers instead of low copy numbers, resulting in a more than 25% increase in IPTG's and aTc's power-law coefficient

(Fig. S7.19). These experimental results are supported by our minimized biochemical model (Fig. S2.8.1).

Furthermore, in principle, perceptron models can compute with continuous numbers and yield output with non-linear behavior depending on the activation function. If the activation function has a sharp response, a hard classification with two-discrete binary levels (low/high) is observed. When the activation functions are soft such as a sigmoid and rectifiers, other complex computing results can be observed that can not be achieved using digital design. Importantly, the IPTG/aTc regulation of $P_{lacO1/tetO}$ promoter via constitutively expressed LacI and TetR implement a conventional Boolean AND logic gate, but the regulatory topology described in Fig. 1D and then in Fig. 1H, using auto-negative feedback, converts the promoter's operation into a logarithmically analog classifier. This discussion was included in our original submission.

12. Sentences like “the power-law functions compute the weighted inputs” are, to me, confusing. Authors are talking about a positive regulation.

We fixed the sentence to “The power-law function encodes weighted inputs by assigning for each input a particular weight, and the multiplication function aggregates the analog values of the weighted inputs.”

13. P7L16 says “perform different computational tasks by modifying various parameters”. What are those parameters? What tasks?

We agree with reviewer. We fixed the sentence to “A perceptgene with AraC/ P_{BAD} activation can be readily customized to perform different computational tasks (e.g., minimum, maximum, and average) by modifying mainly arabinose concentration (fig. S2.9b)”

14. P16L6 “the best performance was observed for a lower level or arabinose”. According to the picture, I do not agree. Please clarify.

We agree with reviewer. We fixed the sentence to “The best performance was observed for an arabinose level between 0.03125mM and 0.0625mM that corresponds to an intermediate P_{BAD} /AraC weight”

15. How is that Fig4H is a switch?

Binary switches are digitizers for analog signals with low and high output digital states. Trinary switches are advanced digitizers with three digital output levels rather than two levels. Both switches act as analog-to-digital converters.

16. *In P13 authors talk about rows in Table 1. This is so confusing. I do not understand the table.*
We thank the reviewer for pointing this out. We added a detailed description for each row in the Table. The explanation of each column and row computation is included in the table caption.

Reviewer #3 (Remarks to the Author):

1. *The manuscript by Rizik et al. describes a framework for engineering genetic circuits that implement neural-like network computations in E. coli. The circuits are based on the classic ligand-inducible transcription factors LacI, TetR, AraC, LuxR. The authors integrate these transcription factors into small feedback-containing circuits (perceptgenes) that perform logarithmic computations - similar to previous work on analog gene circuit computation. The critical parameters of weight and bias are manipulated via an assortment of ligand concentration, transcription factor, and promoter modifications. Multiple perceptgenes are inter-connected via additional regulated promoters to implement genetic majority computation and analog-to-digital conversion (both systems convert the presence of ligands to gene expression output patterns at steady state). The experimental work is guided and interpreted by theory.*

We appreciate the reviewer's support for the technical aspects of this paper.

2. *The implementation of neural network computations in gene regulatory networks is a fascinating result. The major limitations of the manuscript are that the authors rely upon idiosyncratic observations about the performance of their gene regulatory parts to implement critical processes such as the tuning of weights and biases that dictate the function of the networks, and the general design principles by which one would vary these variables in genetic circuits is not clear. Thus, most of the future advancements on this work will likely require researchers 'starting from scratch' with other classes of parts with more programmable properties and re-conceiving of approaches to tune weights, biases, etc.*

We thank the reviewer for the valuable feedback, which helped us further clarify and emphasize the contributions of our work. In our response below, we provided comprehensive responses to the points raised by the reviewer above.

In this study, the design of neuromorphic gene circuits is based on the following steps:

- 1. Solve linear equations to determine the topology and initial weights and biases (Fig. 3 and 4)**
- 2. Build the circuit and measure the response**
- 3. Optimize the circuit performance by using small molecules (such as Arabinose in Fig. 1, 2, 3 and 4, and AHL in Fig. 5) and applying gradient descent and backpropagation algorithms (Fig. 4).**

Regarding the biological implementation, we provided (Supplementary information section 8):

(1) A comprehensive study of principles and rules for designing neuromorphic genetic circuits, describing how we can use gene circuits and biochemical reactions to design and build neuromorphic circuits in living cells.

(2) Several biological factors and mechanisms for modulating the weights and biases of a genetic perceptgene.

(3) Computational tools that combine predictive biophysical models with optimization algorithms to design synthetic DNA with desired features.

Section 8 in Supplementary Information includes analytical, simulation, and experimental results. Further details describing the above three points are provided in the following comments.

I also have a number of other concerns. Specific comments below:

Major comments

3. A major limitation of the paper is that the weights of inputs cannot be systematically engineered.

Rather, the authors usually empirically determine the weights of commonly used gene regulatory parts, or small libraries or random mutants derived from those parts, and then select parts with appropriate weights to implement specific neural computations.

We appreciate this concern and agree with the reviewer that systematically engineering the weights and biases could further enhance the computational capabilities of genetically encoded neural networks. In our original submission, we used the Manhattan learning rule, which updates the weight to the nearest available value from a library we created instead of computing the exact weight. Based on the sign of the partial derivative, the algorithm points toward the available higher or lower weight. This technique determines only the direction of changing the weight (to increase or decrease) without the need to obtain precise weight values. In the case of altering m_1 , we increase the Arabinose concentration until minimizing the error. In the case of altering n_1 , we started by introducing random mutations to the first two base pairs of the LuxR operator and only then we moved to the first three base pairs. Overall, we claim it is not essential to optimize the circuit performance with an error equal to zero.

Furthermore, the difficulties of engineering design parameters for neuromorphic genetic circuits are roughly the same as existing digital and analog circuits in living cells. To the best of our knowledge, designing digital circuits requires building vast libraries of synthetic parts in the optimization process.

The process of fine-tuning design parameters of the three computational platforms, analog/digital/neuromorphic, begins with a hypothesis of modulating a regulatory element's dosage response (e.g., transfer function). This process often relies on existing knowledge by using what has already been demonstrated in the literature and existing approaches or developing completely new methods. After implementing the circuit modifications, the new transfer functions are measured and evaluated to determine their design parameters.

In addition, different articles have demonstrated the ability to design various properties of engineered gene circuits that are relevant to our neuromorphic circuit engineering efforts using computational tools. These tools combine predictive biophysical models with computational optimization algorithms to design synthetic DNAs with desired features. For examples (this discussion is added to the Supplementary Information, Section 8):

- The Ribosome Binding Site Calculator is a tool that predicts the binding affinity of a Ribosome and synthetic binding sites in *Escherichia coli*. It enables rational control over transcription factor expression levels Salis *et al.* 2009). This tool can be useful for programming perceptgene bias, which is directly affected by the translation rate.
- The RNA polymerase Binding Site Calculator is a tool that predicts the binding affinity of a RNA polymerase and synthetic promoters in *Escherichia coli*. It enables rational control over transcription factor expression levels (La Fleu *et al.* 2021). This tool can be useful for programming perceptgene bias, which is directly affected by the translation rate.

In general, weights and biases in our neuromorphic circuits are determined by several factors that include, but are not limited to:

- (1) Hill coefficients of small-molecule inducers serving as perceptgene inputs (Fig. 1 and 2),
- (2) number and sequence of transcription factor binding sites (Fig. 1 and 3),
- (3) circuit topology such as the regulation of negative feedback (Fig. 5G, fig. S2.8.1, fig. S8.3),
- (4) circuit topology such as the regulation of incoherent feedforward strength (fig. S8.5),
- (5) transcription factors that competitively inhibit expression via steric hindrance (fig. S8.6),
- (6) transcription factor sequestration via protein-protein interactions (Fig. 5, fig. S8.7),
- (7) operator sequence that controls binding affinity of transcription factor (fig. S8.8 and S8.10),
- (8) protein structure (*e.g.*, dimerization and cooperativity),

4. Furthermore, inducer (arabinose) concentration is used to alter the weight of AraC. This appears to work due to the biochemical details of the AraC/P_{bad} system and is not likely to generalize to other inducible promoters. implemented using the approach in the paper. The author should add a discussion of these limitations of their approach.

We thank the reviewer for pointing this out. In the revised manuscript, we provide a new experiment of a two-input (TPTG, aTc) perceptgene, in which its internal weight is programmed by an AHL molecule instead of Arabinose (Fig. 5). In our P_{lux} construct, the absence of AHL leads to the expression of basal level, and the presence of AHL activates the P_{lux} promoter for high levels. In contrast to the AraC/P_{BAD} system, the absence of Arabinose represses the promoter activity for very low levels, and the presence of Arabinose activates the promoter for high levels. The new experiment

utilizes protein sequestration to control the weights of a two-input perceptgene circuit, demonstrating our approach's versatility for altering weights and biases. Specifically, we selected ExsD to shunt ExsA from binding P_{exs} promoter and activating GFP expression. In our design, we controlled ExsD by AHL: (1) when AHL is low, the internal weight is high, which leads to an output GFP signal with an OR-logic gate manner (Fig. 5E). (2) When AHL is high, the internal weight is low, which leads to an output GFP signal with an AND-logic gate manner (Fig. 5F). To improve the accuracy of the OR circuit and get distinct low and high outputs (Fig. 5G), we increased the input weights by enhancing the strength of the auto-negative feedback loops. In the modified design, we constructed the auto-negative feedback loops of P_{lacO1} and P_{tetO} in medium plasmid copy numbers instead of low copy numbers, which resulted in a more than 25% increase in IPTG's and aTc's power-law coefficient (Fig. S7.19). These experimental results are supported by our minimized biochemical model (Fig. S2.8.1).

5. Finally, arabinose also alters the bias of AraC, which confounds the experiments. Overall, these limitations appear to significantly constrain the breadth of neural network operations that can be

Weight is generally determined by calculating the log domain slope of the regulated promoter's dosage response curve, which is quantified by Hill coefficients. The bias is usually determined by the ratio between the maximum protein expression level and the dissociation constant of the transcription factor binding to DNA (Methods, BOX I). Generally, using one of the mechanisms described above to modulate the weight can also influence the bias and vice versa; however, our computational backpropagation algorithm already compensates for the bias change. During the design process of the neuromorphic gene circuit, we first characterize all the synthetic parts, including the relation between the weights and the bias. Then, we incorporate this relation in conjunction with the chain rule formula to determine the bias derivatives with respect to weight changes.

Transcription factor sequestration is another example that demonstrates how we can modify the weight while maintaining the bias at the same level (Fig. 5C). This circuit is a part of the system described above, where we used ExsD to shunt ExsA from binding P_{exs} promoter which expresses GFP. We used the IPTG/LacI to regulate the expression of ExsA and AHL to induce the expression of anti-activator ExsD through regulation of P_{luxTGT}/LuxR. The resultant IPTG-GFP transfer functions for various AHL concentrations indicated that this sequestration significantly decreased the input IPTG's Hill-coefficient and hence modulated its internal weight (Fig. 5C). Also, the experiment results of IPTG-GFP revealed an increase in the dissociation constant of IPTG and a decrease in the maximum fold change of P_{exs} (Fig. 5C). However, since the bias is inversely

proportional to the dissociation constant (Methods, BOX I), and the maximum fold change is proportional to the threshold of the perceptgene's activation function, we concluded that titrating AHL affects mainly the perceptgene's internal weight.

Below we describe several published methods which allow us to control the maximum protein expression level continuously (e.g. by promoter strength, RBA strength, codon usage, etc...). In the original manuscript submission, we showed that the dissociation constant can be modulated via transcriptional interference (Fig. 4). We showed that weights can be programmed using circuit topology. We demonstrated this experimentally via negative feedback (fig. S8.3) and incoherent feedforward loops (fig. S8.5, Fig. 5G). We believe that our experimental and computation results, combined with what is already known in the literature, provide a solid foundation for effectively realizing modulating weights and bias for neuromorphic circuits.

In fig. S8.3 we describe small molecule control of negative feedback regulation via transcription factor activation of repressor. This design allows us to continuously program the weight of IPTG by changing the level of AHL. Input IPTG regulates promoter P_{lacO} 's expression of LuxR. The LuxR/AHL complex regulates LacI levels, which represses promoter P_{lacO} , creating a negative feedback loop. The negative feedback loop strength, which is controlled by AHL, determines the IPTG input weight.

In fig. S8.5, we demonstrated experimentally small molecule control of transcription factor competitive inhibition via binding to an output promoter. This design allows us to modulate the weight of input AHL continuously by changing the aTc level. The input AHL binds LuxR and forms a complex that induces expression of activator (AraC) and repressor (LacI), which combine to regulate GFP output, resulting in an incoherent feed-forward loop (iFFL, fig. S8.5b). Small molecule aTc controls LacI expression via de-repression of TetR, which in turn affects the overall AHL-GFP transfer function. We showed the resulting input weight as a function of aTc relevant for the input dynamic range (fig. S8.5e).

In fig. S8.6, we showed competitive inhibition of gene activation via steric hindrance binding of DNA that is tuned by the DNA binding location of the dCas9/sgRNA complex. This design allows us to program bias continuously by choosing different sgRNA sequences. In principle, single base and multi-base sgRNA/DNA binding mismatches can be introduced to further modulate this bias. The complex sgRNA-dCas9 binds the araC operator, preventing the AraC/Arabinose complex from activating promoter P_{BAD} . The affinity of dCas9/sgRNA to its binding site and ability to sterically hinder transcription factor binding the promoter control the binding dissociation constant of AraC/Arabinose to P_{BAD} . These factors, and hence perceptgene bias, can be readily controlled by building a library of sgRNA sequences.

In fig. S8.7, we demonstrated small molecule control of transcription factor sequestration via protein-protein interactions. This design allows us to continuously program perceptgene bias using two heterologous proteins (ExsA and ExsD) where ExsA transcriptional activator is sequestered by ExsD into an inactive complex. AraC/Arabinose regulates expression of ExsA that activates GFP output expression. aTc induces expression of anti-activator ExsD which inhibits ExsA gene activation. Hence, the extent of the ExsA/ExsD protein-protein interaction and resultant perceptgene bias is controlled by aTc (fig. S8.7e).

Fig. S8.8 describes modulation of transcription factor LuxR's DNA binding affinity via changes in Lux operator sequence. We introduced seven random mutations into the first four nucleotides of the LuxR binding site and incorporated these into an open loop circuit topology where constitutive LuxR activates expression of GFP from lux promoter variants. Our experimentally measured Hill-coefficient values range between 0.4 and 1.0 and the computed weights range between 0.25 and 0.8. We then incorporated these lux operator mutants into a Lux response circuit with positive feedback regulation (fig. S8.10), and obtained Hill-coefficient values ranging between 1.1 and 2 and computed weights between 0.75 and 1.7.

In addition to our own experimental data, previous articles have also demonstrated the ability to modulate various properties of engineered gene circuits that are relevant to our neuromorphic circuit engineering efforts. The engineered libraries of genetic device variants described briefly below could be used in our neuromorphic approach to obtain essentially continuous modulation of weights and biases:

- The Ribosome Binding Site Calculator is a tool that predicts the binding affinity of a Ribosome and synthetic binding sites in *Escherichia coli*. It enables rational control over transcription factor expression levels (Salis *et al.* 2009). This tool can be useful for programming perceptgene bias, which is directly affected by the translation rate.
- The RNA polymerase Binding Site Calculator is a tool that predicts the binding affinity of a RNA polymerase and synthetic promoters in *Escherichia coli*. It enables rational control over transcription factor expression levels (La Fleu *et al.* 2021). This tool can be useful for programming perceptgene bias, which is directly affected by the translation rate.
- The Anderson synthetic promoter library includes more than 30 characterized promoters with variable strength of approximately 100 fold between the weakest and strongest (<http://parts.igem.org/Promoters/Catalog/Anderson>). This tool can also be useful for programming bias, which is directly affected by the transcription rate. Another synthetic promoter library was also published around the same time (Alper *et al* 2005).

- Our previously published TALER library includes 26 programmed transcriptional repressors that bind synthetic combinatorial promoters in mammalian cells (Li *et al.* 2015). With TALE modular protein construction, any DNA sequence can be targeted, leading to an essentially limitless design search space (with a practical length of anywhere between 14 and 26 DNA bases for TALER binding). The library elements have an approximately 2 orders of magnitude difference in repression folds from around 20 to greater than 10^3 leading to different Hill coefficients and hence different input weights. Promoter engineering by inclusion of two versus four TALER binding sites increased fold repression by five and ten fold for TALER21 and TALER14 respectively.
- The Voigt repressor library includes 16 orthogonal TetR-family repressors and their cognate promoters. Each repressor/promoter pair's transfer function has been characterized. The measured Hill coefficients range between 1.5 and 6.5, with fold changes between 1-2 orders of magnitudes (Stanton *et al.* 2014). This tool can be useful for programming input weights similar to fig. S8.8.
- A recent effort in *Escherichia coli* has demonstrated several inducible synthetic promoters with varying ligand-promoter activity transfer functions. *The synthetic promoters are regulated by TtgR, PmeR and NalC and are induced by phloretin, Naringenin, and PCP, receptively (Liu et al 2019).* This tool can be useful for programming input weights similar to fig. S8.8.
- The Riboswitch Binding Sequence Calculator predicts ligand induced gene activation of riboswitch sequences using a physics based model. Then, computational design with this tool is used to create a library of 62 different synthetic riboswitches with activation fold of up to 383x (Borujeni *et al.* 2015). This tool can be useful for programming weight and bias.
- A library of LuxR transcription factors were developed (Zucca *et al.* 2014). AHL-dependent transcriptional activation can be selected to meet design specifications. This tool can be useful for programming input weight similar to fig. S8.8.
- A library with 238 elements providing tunable control of protein degradation in bacteria was described previously (Cameron *et al.* 2014). This library is useful for programming bias, which is directly affected by the protein half-life.
- A library of antisense regulated promoters was developed (Brophy *et al* 2016). Every member of the library includes a target gene that is regulated by a repressor and another promoter oriented in the opposite direction to the target gene which constitutively expresses antisense RNA. The library includes 5,668 promoter combinations used to modulate the regulatory effect of three repressors (PhIF, SrpR, and TarA). Such a design can reliably tune gene

expression levels and control the effective dissociation constant of small molecules via transcription factor binding to the promoter, and hence aids in programming bias.

Other methods to alter the dosage response curves of genetic regulation elements have also been published, and these could also be used to modulate weight and bias in neuromorphic circuits:

- Landry *et al.* 2018 developed a two-component signaling system that can dynamically tune the dissociation constant of small molecules. This system can be used to control bias.
- Segall-Shapiro *et al.* 2014 split T7 proteins into several parts and changed cooperativity. This method can be used to control input weight.
- Morel *et al.* 2016 introduced extra binding sites into promoters and changed cooperativity. This method can be used to control input weight.

In addition, we analyzed the properties of common synthetic biological parts, including weights for some of the parts used in this manuscript (fig. S8.11a) and Hill coefficients for devices that were previously published (fig. S8.11b) providing another source of parts with desired weights and Hill-coefficients for small molecules and transcription factors. We discussed these points in the main narrative, Page 24, lines 4 to 6.

6. The fluorescent protein reporter data are reported in a highly inconsistent manner throughout the paper. For example, the data demonstrating the perceptgene power law and multiplication sub-circuit function is plotted in normalized units in Figures 1C and D, but in non-normalized units in Figures S9.1 and S9.2. The lack of standardization of units makes it difficult to compare the performance of different systems or the same system in different experiments or genetic contexts.

We thank the reviewer for pointing this out. In the revised manuscript, we plotted Fig. 1, Fig. 2, and Fig. 9 with the same representation using three-dimension space.

7. The authors claim that the P_{bad} promoter exhibits a log-transformed sigmoidal response (page 9, paragraph 2 and Fig. S2.9). Based on the description and model of perceptgene function that the authors provide elsewhere in the manuscript, one would assume that the relevant response of P_{bad} is its response to the concentration of AraC. However, the experiment performed in Fig. S2.9 does not provide the response of P_{bad} to AraC concentrations. In particular, the authors use AHL to induce expression of a degradation tagged version of AraC from a positive autoregulatory promoter (Fig. S2.9a). The relationship between AHL and AraC levels is not measured, and it should not be assumed that this relationship is linear, especially due to the positive autoregulation and degradation tag in this characterization circuit. On a related note, how do the authors arrive at log(AraC) values that they use to show negative rectifier function

in Fig. S3.1? It does not appear they have the necessary data to conclude negative rectifier function.

We thank the reviewer for pointing this out. The relation between AHL and AraC is measured and determined in Fig. S2.13. In the experiment shown in Fig. S2.13, we used the same positive feedback circuit that controls AraC in Fig. S2.29 to express GFP. Thus, the level of AraC in Fig. S2.9 can be obtained by the GFP signal from Fig. S2.13C. We added this description in the caption of Fig. S2.9d. To get better insight into the P_{BAD} promoter, we provided a new model (fig. S2.11 A) that captures the influence of low and high arabinose concentrations on the P_{BAD} promoter. Based on our model, P_{BAD} activity induced with low Arabinose can be approximated by a negative rectifier, and with high Arabinose can be approximated by a positive rectifier. This conclusion is consistent with our results from Fig. S3.1.

8. The origin of the perceptgene model (Supplementary Information, Section 1 - Box 1) is not described. Thus, it is not clear how this model facilitates the design or understanding of the performance of a perceptgene. For example, why is perceptgene bias determined by "translation and transcription rates, mRNA/protein half-lives, rate of cell growth, binding affinities in protein-protein or protein-DNA reactions".

We thank the reviewer for the suggestion. First, we moved BOX I to Methods (Methods, BOX I). Second, we added additional content describing the required steps to develop the perceptgene abstract model. We started with the formalization of a hybrid promoter having multiple inputs, and then we defined an ordinary differential equation for gene expression. Finally, we emphasize the role of translation/transcription rates, mRNA half-life through the production rate, protein half-life, and cell growth rate.

9. How do such a large number of disparate process combine to result in a bias?

A similar process is also valid in digital design. The binding dissociation constant of a transcription factor to DNA behaves as a threshold, which activates the logic gate from low to high. Also, the wiring of one circuit's output to another's input leads to 'loading' interactions that degrade overall function and prevent modularity of digital circuits. Thus, in the case of digital gene circuits, the maximum expression level of a transcription factor drives the next logic gate, and the dissociation constant behaves as a load (e.g., switching threshold). As a result, the ratio between the maximum expression level of a transcription factor and its binding dissociation constant to DNA represents the ratio between the driver and the load in digital circuits (Sarpeshkar *et al.*, 2016), while this ratio represents a bias in neuromorphic circuits.

Furthermore, the unique process of the bias that involves many disparate processes dictates the operation with low expression levels in neuromorphic gene circuits, mainly to maintain low bias levels. In contrast, this unique process dictates digital circuits to operate with high expression levels to prevent loading interactions.

10. How would one design or modify a perceptgene to modify the bias given this complex entanglement of processes?

In this manuscript, we used several biological factors to manage the bias. Among these are weak ribosome binding sites, weak RNA polymerase binding sites, fusion transcription factors with ssrA degradation tags, and encoding promoters and proteins in different plasmid copy numbers. Our original submission included a chapter named “Design principles of neuromorphic gene circuits” in the supplementary information, which theoretically and experimentally presents several biological mechanisms to program the bias.

The bias is controlled by modifying the dissociation constant, translation and transcription rates, and protein half-lives. We showed that the dissociation constant could be modulated via transcriptional interference (Fig. 4). Furthermore, in fig. S8.6, we showed competitive inhibition of gene activation via steric hindrance binding of DNA that is tuned by the DNA binding location of the dCas9/sgRNA complex. This design allows us to continuously program the bias by choosing different sgRNA sequences. In principle, single base and multi-base sgRNA/DNA binding mismatches can be introduced to further modulate this bias. The sgRNA-dCas9 complex binds the araC operator, preventing the AraC/Arabinose complex from activating promoter P_{BAD} . The binding affinity of dCas9-sgRNA complex to P_{BAD} controls the Arabinose- P_{BAD} characteristic. These factors, and hence perceptgene bias, can be readily controlled by building a library of sgRNA sequences. In fig. S8.7, we demonstrated small molecule control of transcription factor sequestration via protein-protein interactions. This design allows us to continuously program perceptgene bias using two heterologous proteins (ExsA and ExsD) where ExsA transcriptional activator is sequestered by ExsD into an inactive complex. AraC/Arabinose regulates the expression of ExsA that activates GFP output expression. aTc induces the expression of anti-activator ExsD, which inhibits ExsA gene activation. Hence, the extent of the ExsA/ExsD protein-protein interaction and resultant perceptgene bias is controlled by aTc (fig. S8.7e). In addition, as discussed above, we can use several computational tools to control bias levels, such as the Ribosome Binding Site Calculator (Salis *et al.*, 2009) and the RNA polymerase Binding Site Calculator (La Fleu *et al.*, 2021).

11. The description of the perceptgene model is poor, making it difficult to follow. For example, in describing equation (1) the authors state "Where X_i is the input concentration, K_{mi} is dissociation constant, or input dynamic range or normalization". Is K_{mi} a dissociation constant? If so, of what? Or is it an input dynamic range? Or is it a normalization term?

We thank the reviewer for pointing this out. In the revised manuscript, we added a further description of the perceptgene abstract model in BOX I (Methods, BOX I). We fixed the definitions of the parameters including K_m .

12. The scales and units differ between the experimental and simulated sub-circuit characterization data in Figures 2B and C. This makes it difficult to compare experimental and simulation results. The authors should account for these discrepancies.

In the revised manuscript, we represent Fig. 2 as a three-dimension plot. This new representation allows better comparison between the experimental and simulation results.

13. The analog-to-digital converter data (Figure 4) is a strength of the paper. I believe that this result shows the implications of the framework in a particularly compelling way.

We are grateful for the kind words and support for this work.

14. On page 21, the authors seem to imply that gene regulatory parts with steep transfer functions (high Hill coefficients) would be beneficial to their framework. However, if or why this is the case is not made clear. The authors should clarify this section of the discussion and enumerate the benefits of such digital-like parts if they would indeed be beneficial.

We thank the reviewer for the suggestion. Here, we raise several points that discuss soft classification versus hard classification.

- **Figures 1, 2, and 3 demonstrated perceptgene networks with soft classification using a sigmoid function, rectifier function, and linear function. To observe a hard classification response with an output that has only two-discrete binary states, we can connect the final layer of the computational networks with an activation function with a sharp response (steep input-output transfer function). For example, we can replace the GFP of the circuit in Fig. 3B with recombinase protein [P Siuti *et al.* 2013] to obtain a majority circuit with clear low and high output signals. We used the "soft" majority function in the revised manuscript to highlight that our majority is smooth. We expanded the discussion section to include this point (page 21, lines 20-22). A further description for implementing a hard majority using recombinase is provided in the supplementary information section 5, Fig. S5.10.**

- Furthermore, in computation and signal processing, there are several advantages to using soft classification rather than hard classification. For example, our theoretical analysis showed that the three-input majority function can be implemented with a single layer of a step activation function (e.g., the three input weights are equal to one, and the threshold equals 1.5). However, implementing an activation function (e.g., a promoter with three different binding sites) with three inputs in living cells and obtaining the desired parameters (weights and bias) is often challenging. Alternatively, we initially thought to implement a three-input majority function by cascading two perceptrons, each having two inputs. From our analysis of such a network, only perceptrons with a sigmoid activation function can satisfy the mathematical constraints on the design parameters to implement the three-input majority function. Therefore, the only feasible way to build a genetic circuit in *E. coli* that computes the majority of three inputs is by using two layers of a two-input sigmoid activation function, as was implemented in this study (Fig. 3). We added this discussion and the theoretical analysis to the Supplementary Information, Section 5, Table S5.3, and Table S5.4.
- In addition, training artificial neural networks with step functions is challenging. First, step functions are non-differentiable and can not be used in the backpropagation algorithm. Second, physical systems with step functions are not stable during the training process. The step function is generally used as a final layer for decision-making capability and not as a hidden layer in the middle of the network.

Importantly, the 2-bit analog-to-digital converter in Fig. 4 and the 2-input logic gates in Fig. 5 demonstrated perceptgene networks with hard classification.

15. The authors state: "We anticipate that the new framework described here constitutes a first step towards implementing supervised machine learning optimization algorithms in individual living cells". This is certainly an intriguing claim. However, the implications of achieving supervised learning in cells, and the technical limitations that must be overcome are not described. This information should be added, or the statement should be removed.

We thank the reviewer for pointing this out. We modified the sentence and added a narrative that describes our study more accurately. "We anticipate that the new framework described here constitutes a first step towards optimizing gene circuit design by applying data-driven algorithms, which determine how one can modify the circuit's design parameters by measuring the circuit's output signals in each step during the optimization process".

16. Why is mCherry used as a reporter gene in Figure 2A-C, when GFP is used throughout the rest of the paper? This peculiar choice compounds with the sporadic use of normalization to make it difficult to interpret the performance of circuits throughout the paper.

We thank the reviewer for the suggestion. In the revised manuscript, we modified the circuit in Fig. 2A to include GFP replacing mCherry. Our experimental results demonstrated that GFP and mCherry have similar behavior. Both circuits are linear at the log scale but measured GFP intensity has a larger fold change than mCherry.

Minor comments:

17. The schematics depicting the gene regulatory network designs are non-standardized and difficult to interpret. For example, in Figure 1, promoters are represented as rightward facing block arrows that occupy the entire height of the depicted DNA. This representation is commonly used to depict open reading frames. Open reading frames are depicted as the absence of any glyph on the DNA, which is also unusual. Additionally, rather than depicting a specific origin of replication or a category of origins or replication using a glyph, the authors write "LCP", "MCP", or "HCP" for low- medium- and high-copy plasmid in text next to the DNA. Furthermore, the style of the depictions changes within the document (e.g. Fig. S.2.2). At a minimum, the schematics should be standardized. I recommend the authors follow current SBOL visual standards.

Fixed in the revised manuscript.

18. The words "LacI binding site" should be replaced with the sequence of the LacI O1 binding site in Fig. S2.5. The label "O1" beneath the LacI binding site region is sufficient to indicate that the sequence is a LacI operator.

We thank the reviewer for pointing this out. This comment is fixed

19. What is the relevance of the fact that the perceptgene responses are stable between 4.5 and 10 hours? Do they fail after 10 hours? In any case, this number is likely to be limited by batch culture growth conditions and adding the inducers at only one time point.

We thank the reviewer for pointing this out. The dynamics of our circuits are mainly determined by the characteristics of the synthetic parts and the regulatory topologies. We used synthetic parts that have been extensively characterized in the literature. The regulatory topologies that govern the behavior of our circuits include cascades, feed-forward, and feedback motifs – again, motifs that occur frequently in synthetic biology. We expect that the dynamics of our neuromorphic circuits are roughly the same as existing digital and analog circuits using similar synthetic parts and motifs

(Tasmir *et al.* 2011, Daniel *et al.* 2013), e.g., response times in a few hours. We did not test the circuit beyond 10 hours.

Reviewers' Comments:

Reviewer #2:

Remarks to the Author:

I honestly think that authors did a very good job in addressing my concerns. Analog computing will definitely (in my opinion, of course!) lead to more reliable genetic circuits, as these principles match the internal workings of living cells. Congratulations on your work.

I have no further concerns with the manuscript. My only remark would be to encourage authors to make all relevant data public (code for simulations, materials, protocols, experimental info...) if at all possible.

Reviewer #3:

Remarks to the Author:

I thank the authors for carefully considering and responding to the the reviewer comments and making the requested changes to improve the manuscript. I believe the manuscript is now sufficient for publication in Nature Communications.

Synthetic neuromorphic computing in living cells
Point-by-point response to the reviewers' comments

September 11, 2022

We thank all reviewers for the comments. *Reviewer comments are italicized.* **Our answers are in bold.**

Reviewer #2 (Remarks to the Author):

I honestly think that authors did a very good job in addressing my concerns. Analog computing will definitely (in my opinion, of course!) lead to more reliable genetic circuits, as these principles match the internal workings of living cells. Congratulations on your work
We are grateful for the kind words and support for this work.

I have no further concerns with the manuscript. My only remark would be to encourage authors to make all relevant data public (code for simulations, materials, protocols, experimental info...) if at all possible.

We have included Source Data excel files, which contain fluorescence signals for all the figures in the manuscript. The source data of each figure is represented by a separate Excel file, and each sub-figure is represented by a separate sheet. Furthermore, we have uploaded data, including measured signals and FACS, to a GitHub and also for the SI. We have included the Matlab codes used for the simulations. We have described protocols and materials used in the Methods section.

The GitHub repository contains: (1) Codes, (2) Experimental data (FACS data), and Source data (Excel files for the raw data)

GitHub ULR: <https://github.com/LSB2/2022-Synthetic-neuromorphic-computing-in-living-cells/tree/0.1.0>

Zenodo ULR: <https://zenodo.org/record/7040614#.YxmnkXbP2Uk>

DOI: [10.5281/zenodo.4682962](https://doi.org/10.5281/zenodo.4682962)

Reviewer #3 (Remarks to the Author):

I thank the authors for carefully considering and responding to the reviewer comments and making the requested changes to improve the manuscript. I believe the manuscript is now sufficient for publication in Nature Communication

We are grateful for the kind words and support for this work.